# Predictors of maternal HIV acquisition during pregnancy and lactation in sub-Saharan Africa: A systematic review and narrative synthesis

Lauren A. Graybill[1]*, Benjamin H. Chi[2], Twaambo E. Hamoonga[3], Margaret Kasaro[2,4], Jasmine N. Hodges[5], Brian D. Richardson[6], Jennifer S. Bissram[7], Friday Saidi[8], Katie R. Mollan[9], Kellie Freeborn[2], Nora E. Rosenberg[10], Kimberly A. Powers[9], Wilbroad Mutale[11]

1 Institute for Global Health and Infectious Diseases, University of North Carolina at Chapel Hill, Chapel Hill, NC, United States of America, 2 Department of Obstetrics and Gynecology, University of North Carolina at Chapel Hill, Chapel Hill, NC, United States of America, 3 Department of Population Studies and Global Health, University of Zambia, Lusaka, Zambia, 4 UNC Global Projects–Zambia, Lusaka, Zambia, 5 Department of Maternal and Child Health, University of North Carolina at Chapel Hill, Chapel Hill, NC, United States of America, 6 Department of Biostatistics, University of North Carolina at Chapel Hill, Chapel Hill, NC, United States of America, 7 Health Sciences Library, University of North Carolina at Chapel Hill, Chapel Hill, NC, United States of America, 8 UNC Project Malawi, Lilongwe, Malawi, 9 Department of Epidemiology, University of North Carolina at Chapel Hill, Chapel Hill, NC, United States of America, 10 Department of Health Behavior, University of North Carolina at Chapel Hill, Chapel Hill, NC, United States of America, 11 Department of Health Policy and Systems, University of Zambia, Lusaka, Zambia

* lag111@email.unc.edu

**Data Availability Statement:** All relevant data are within the manuscript and its Supporting Information files.

## Abstract

### Objectives

To eliminate vertical transmission of HIV, global institutions recommend using a risk-guided approach for HIV prevention services in antenatal and postnatal settings. Identifying predictors of maternal HIV acquisition can inform the development of risk-guided approaches, but individual studies of predictors can have limited power and generalizability.

### Methods

We conducted a systematic review and narrative synthesis to identify common predictors of maternal HIV acquisition in sub-Saharan Africa (SSA). We searched four databases for full-text articles that estimated associations between at least one predictor and risk of HIV acquisition among pregnant and/or lactating women (PLW) in SSA. We restricted our synthesis to predictors assessed in at least four study populations. For these predictors, we summarized how each predictor was defined and used vote counting and descriptive statistics to characterize overall trends.

### Results

We identified 26 eligible publications that summarized results from 24 unique studies. Studies were implemented in 12 countries between 1988 and 2021 and enrolled a total of

**Funding:** This study was funded by the National Institute of Allergy and Infectious Diseases (NIAID) through award R01 AI131060. Additional investigator, trainee, and administrative support was provided by NIAID (T32 AI007001, K24 AI120796, P30 AI050410, R01 AI157859), and the Fogarty International Center (D43 TW009340, D43 TW010558). The funders had no role in study design, data collection and analysis, decision to publish, or preparation of the manuscript.

**Competing interests:** Drs. Graybill and Chi received consulting fees from UNICEF. This does not alter our adherence to PLOS ONE policies on sharing data and materials. All other authors reported no conflicts of interest.

164,480 PLW at risk of acquiring HIV. Of the 66 predictors evaluated, 16 met our inclusion criteria. Estimated associations tended to be imprecise and variability in how predictors were measured precluded meta-analyses. We observed trends towards a higher risk of maternal HIV acquisition among young women and women who reported early coital debut, multiple partnerships, sexually transmitted infections, being unaware of partner HIV status, or having a partner living with HIV. In most studies, PLW in stable, monogamous relationships experienced a lower risk of acquiring HIV than those who were single, separated, or in a polygynous marriage. HIV risk perception, condom use, and vaginal drying were also commonly associated with HIV acquisition risk.

## Conclusions

In our systematic review and narrative synthesis, we identified several easily measured characteristics that were associated with HIV acquisition among PLW in multiple study populations across SSA. Such findings can support the development and refinement of risk-guided approaches for HIV prevention in the region.

## Introduction

In 2020, an estimated 120,000 pregnant and/or lactating women (PLW) acquired HIV across the 21 UNAIDS focus countries [1]. HIV acquisition during pregnancy or lactation has important implications for maternal health and accounts for a large—and growing—proportion of new pediatric HIV infections in the Option B+ era [1, 2]. To reduce risk of maternal HIV acquisition and support efforts towards elimination of vertical transmission, the World Health Organization (WHO) recommends offering all PLW who initially test HIV-negative in high-burden antenatal or postnatal settings a comprehensive package of HIV prevention services, including repeat maternal HIV testing, partner HIV testing and referrals, partner referral for voluntary medical male circumcision, screening and treatment for sexually transmitted infections, condom promotion and provision, and risk reduction counseling [3]. Additionally, the WHO endorses the provision of pre-exposure prophylaxis (PrEP) to PLW at substantial risk of HIV acquisition [3], and to those who request it [4].

Meta-analyses of studies conducted in SSA have reported high average maternal HIV incidence rates during pregnancy and lactation [5, 6]. Biological and behavioral factors contribute to the high risk of HIV acquisition during these periods. Hormone-associated changes in the female genital tract (e.g., changes in vaginal epithelia thickness), and suppression of the female immune response are believed to increase biological susceptibility to HIV during pregnancy [7, 8], while behavioral changes during pregnancy (e.g. reductions in condom use) may increase exposure to HIV [8–10]. Even in populations with high average HIV risk, however, individual risk varies according to individual- and partner-level behaviors and characteristics and population-level attributes [4, 11].

To enhance cost-effectiveness and public health impact, the WHO and UNAIDS recommend a risk-guided approach to HIV prevention efforts in antenatal and postnatal settings [3, 4, 11]. Identifying subgroups of PLW at particularly high risk of acquiring HIV is therefore important for optimizing implementation of prevention of mother-to-child transmission (PMTCT) programmes in the region. We conducted a systematic review and narrative synthesis to characterize common predictors of maternal HIV acquisition in SSA, with the goal of

informing future efforts to effectively identify and prioritize PLW at greatest risk of acquiring HIV in the region [12].

## Methods

This systematic review is registered with PROSPERO (CRD42017079577) and follows the Preferred Reporting Items for Systematic Reviews and Meta-Analysis Guidelines [13]. We searched PubMed, Embase, PsycInfo, and the Cochrane Library databases from January 1, 1980 to March 31, 2024 for studies evaluating predictors of maternal HIV acquisition during pregnancy and/or lactation in SSA. This search strategy was developed in consultation with a research librarian at the University of North Carolina at Chapel Hill, and included terms related to pregnancy and lactation, sub-Saharan Africa, HIV, and risk factors (S1 Table). We downloaded search results to EndNote (Clarivate, Philadelphia, PA, USA), where we removed duplicate publications. We then uploaded the EndNote library to Covidence (Veritas Health Innovation, Melbourne, Australia) for publication screening.

In duplicate, two reviewers independently screened titles and abstracts to identify publications that broadly referenced either maternal HIV acquisition or predictors of HIV acquisition. Following screening, two reviewers independently assessed full-text publications to identify publications with estimates of, or sufficient information to derive, a crude (unadjusted) association between at least one predictor and HIV acquisition among PLW in SSA. We excluded non-English publications, conference abstracts, secondary research publications (e.g. modeling studies and literature reviews), publications that did not measure incident HIV among PLW in SSA, and publications that only reported adjusted estimates of association since such estimates are not easily interpreted, combined, or compared [14, 15]. Prior to abstract screening and full-text review, we conducted a calibration exercise to ensure inter-reviewer reliability. Throughout the review process, the review team met regularly to discuss and resolve disagreements on inclusion/exclusion decisions.

For each included publication, we evaluated risk of bias using the Quality in Prognosis Studies (QUIPS) tool [16], and used a standardized form to record information on study characteristics, definitions of HIV incidence during pregnancy and/or lactation, definitions of predictors, and analytical approaches used to estimate associations. Because our analysis focused on crude associations (and thus not causal relationships), we did not consider questions on the QUIPS tool related to confounding when evaluating risk of bias. To facilitate the identification of trends in associations between predictors and risk of HIV acquisition among PLW, we restricted our synthesis to predictors evaluated in at least four unique study populations. For these predictors (hereafter referred to as predictors of interest), we recorded crude measures of association (risk ratios, odds ratios, hazard ratios, or incidence rate ratios) and 95% confidence intervals (CI) from each publication, and calculated confidence limit ratios (CLRs) to quantify precision [17, 18]. The CLR is a standardized measure (upper limit of the CI divided by the lower limit of the CI) that reflects the degree of variability in the estimated association. If a cohort study reported only odds ratios but provided tabulated data, we computed risk ratios and corresponding CIs to improve interpretability. If crude associations or 95% CIs were not reported, we computed appropriate measures using available data. When a predictor was heterogeneously defined across studies and there was no strong evidence that the magnitude or direction of association differed according to definition, we identified a standard set of contrasts to estimate using available data. If more than one publication reported on the same underlying study, we reviewed information from all related publications and reported each unique result. One reviewer (LAG) extracted data from each study and a second reviewer (TEH or JNH) verified results.

Given diversity in the type and definitions of predictors, we synthesized the literature following guidelines for synthesis without meta-analysis [19]. For each predictor, we described how the predictor was measured and operationalized for analyses and we used vote counting to summarize the direction of association between the predictor and maternal HIV acquisition [20]. Following best practices, we classified each estimate as providing evidence of a positive or inverse association according to the direction of the estimate; we did not consider the magnitude or precision of the estimate in the vote counting process [20]. If more than 70% of estimates indicated the same direction of association, we interpreted this as evidence of a trend. For each predictor, we also used descriptive statistics to summarize the magnitude and precision of estimated rate and risk ratios [20]. Because hazards can be interpreted as incidence rates [21], we included hazard ratios in our summary of rate ratios. Likewise, because odds ratios approximate risk ratios when incidence of the outcome is <10% (i.e. the rare disease assumption) [22], a threshold met for estimated HIV incidence among PLW in most prior studies [5], we included odds ratios in our summary of risk ratios.

## Results

Our electronic database search returned 7,274 unique publications, 7,082 of which were excluded during title and abstract screening. Of the remaining 192 publications, 165 were excluded following full-text review and one was excluded following data extraction (Fig 1). A total of 26 publications, reflecting 24 unique study populations, contributed at least one estimate of association to this review (Table 1). Studies were implemented in 12 countries in SSA, with midpoints of implementation falling before 2010 (n = 8), between 2010 and 2014 (n = 9), and after 2014 (n = 7). Of the 164,480 PLW at risk of acquiring HIV during pregnancy or breastfeeding who enrolled in these studies, most lived in southern Africa (n = 150,552), followed by eastern Africa (n = 13,451) and central Africa (n = 477). No studies were conducted in western Africa. The smallest study (implemented in Rwanda between 1988 and 1992) evaluated outcomes among 217 women and the largest study (implemented in Botswana between 2017 and 2021) evaluated outcomes among 86,282 women. The median sample size across all studies was 1,330 (interquartile range: 728–5,044).

A total of 1,402 incident HIV infections were reported across the 24 study populations. Eighty-three percent of cases (n = 1,159) were identified by studies conducted in southern Africa, 17% (n = 234) by studies conducted in eastern Africa, and <1% (n = 9) by studies conducted in central Africa. Incident cases were identified in four ways. Prospective studies—stand-alone (n = 12) or nested within HIV surveillance systems (n = 3)—followed HIV-negative PLW women and retested them at specific intervals to identify incident HIV infections. Six cross-sectional studies enrolled PLW with a documented HIV-negative test result from the index pregnancy and retested them at enrollment—which occurred later in pregnancy or during lactation—to identify new HIV infections. One cross-sectional study enrolled pregnant women who tested negative on a rapid HIV test administered at enrollment and used HIV RNA nucleic acid amplification tests to identify those with acute HIV infections (AHI) at enrollment. In this study, women with AHI were classified as having acquired HIV during pregnancy. Finally, two cross-sectional studies enrolled pregnant women living with HIV and used recent infection testing algorithms (RITA) to identify those who recently acquired HIV. In these studies, women with recent HIV infections were classified as having acquired HIV during pregnancy.

All publications were classified as having moderate or high risk of bias in at least one domain of the QUIPS tool (Table 2). Few publications provided evidence that their study sample was representative of a meaningful target population, and several reported suboptimal

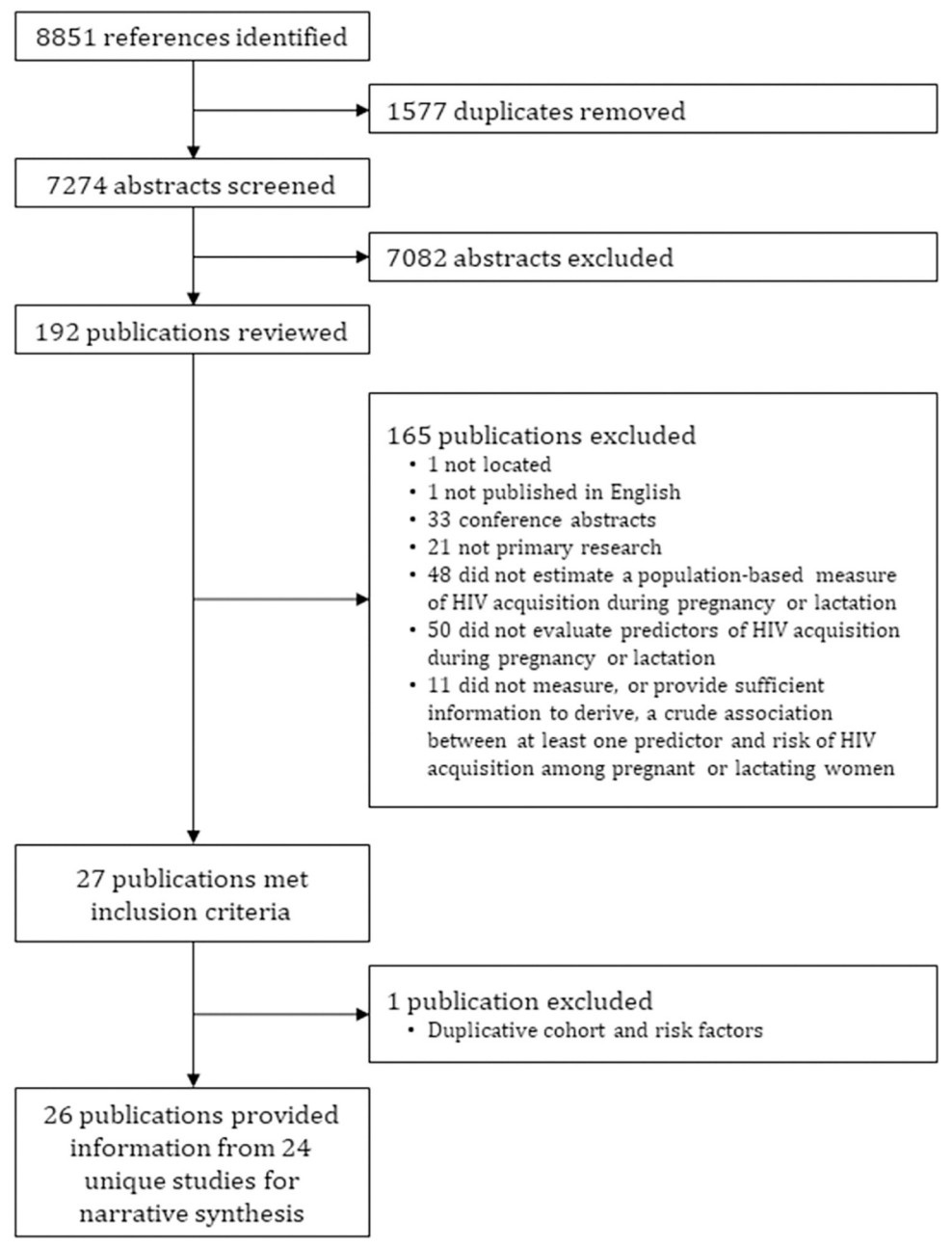

**Fig 1. Study selection flow chart.**

enrollment among eligible women. High attrition was another common source of potential bias, with some publications reporting losses to follow-up that exceeded 50%. Most publications adequately measured predictors and outcomes. Those that did not failed to provide sufficient information on how data were collected or how predictors or outcomes were defined, relied on self-reported data as part of outcome construction, or used RITAs that included part of the pre-conception period in the definition of a recent HIV infection. We note that we did not consider the use of self-reported data to measure predictors a limitation because this approach emulates how such data would be collected in real-world clinical settings. Finally,

**Table 1. Characteristics of included publications.**

| Author (Year) | Country | Years of study implementation | Study design | Risk period | | Number of participants at-risk of acquiring HIV during risk period | Number of participants who acquired HIV during risk period | Number of predictors measured in study |
|---|---|---|---|---|---|---|---|---|
| | | | | **Start** | **End** | | | |
| Objectives included the identification of predictors of maternal HIV acquisition during pregnancy and/or breastfeeding | | | | | | | | |
| Bulterys (1994) | Rwanda | 1991–1993 | Cross-sectional (retesting) study nested within a prospective cohort study | Pregnancy (first HIV test) | ~24 months postpartum | 1,150 | 31 | 15 |
| De Schacht (2014a) | Mozambique | 2008–2011 | Prospective cohort study | Pregnancy (~24 weeks) | Delivery | 1,230 | 14 | 19 |
| De Schacht (2014b) | Mozambique | 2008–2011 | Prospective cohort study | Delivery | ~18 months postpartum | 1,221 | 41 | 21 |
| Dinh (2015) | South Africa | 2011–2012 | Cross-sectional (retesting) study | Pregnancy (first HIV test) | ~8 weeks postpartum | 7,064 | 212 | 18 |
| Egbe (2016) | Cameroon | 2011 | Cross-sectional (retesting) study | Pregnancy (first HIV test) | Pregnancy (second HIV test) *Repeat HIV test occurred three to six months after initial HIV test.* | 477 | 9 | 12 |
| Fatti (2016) | South Africa | 2013–2016 | Prospective cohort study | Pregnancy (~16 weeks) | ~18 months postpartum | 1,356 | 11 | 8 |
| Hira (1990) | Zambia | 1987 | Prospective cohort study | Delivery | ~12 months postpartum | 634 | 16 | 6 |
| Kinuthia (2010) | Kenya | Not reported | Cross-sectional (retesting) study | Pregnancy (first HIV test) | Infant's six-week immunization visit | 2,035 | 53 | 9 |
| Kinuthia (2015) | Kenya | 2011–2013 | Prospective cohort study | Pregnancy (~27 weeks) | ~9 months postpartum | 1,304 | 25 | 21 |
| Machekano (2018) | Lesotho | 2013–2015 | Prospective cohort study | Pregnancy (~27 weeks) | ~24 months postpartum | 850 | 28 | 9 |
| Mbena (2014) | Tanzania | 2013 | Cross-sectional (retesting) study | Pregnancy (first HIV test) | Delivery *A minimum of three months between first test and delivery was required.* | 400 | 21 | 8 |
| Miotti (1994) | Malawi | 1989–1993 | Case-control study nested within a prospective cohort study | Delivery | ~24 months postpartum | 687 | 43 | 9 |
| Mussa (2023) | Botswana | 2017–2021 | Prospective cohort study nested within a surveillance system | Pregnancy (~16 weeks) | Delivery or last HIV test during pregnancy | 86,282 | 223 | 8 |
| Schumann (2020) | Uganda | 2017 | Cross-sectional (retesting) study | Pregnancy (first HIV test) *Test had to occur before the third trimester.* | Delivery | 1,610 | 15 | 30 |
| Taha (1998) | Malawi | 1990–1993 | Prospective cohort study | Delivery | ~30 months postpartum | 1,196 | 27 | 6 |
| Tavengwa (2007) [a] | Zimbabwe | 1997–2001 | Prospective cohort study | Delivery | ~12 months postpartum | 1,890 | 49 | 7 |

*(Continued)*

**Table 1.** (*Continued*)

| Author (Year) | Country | Years of study implementation | Study design | Risk period | | Number of participants at-risk of acquiring HIV during risk period | Number of participants who acquired HIV during risk period | Number of predictors measured in study |
|---|---|---|---|---|---|---|---|---|
| | | | | **Start** | **End** | | | |
| Van de Perre (1991) | Rwanda | 1988–1990 | Case-control study nested within a prospective cohort study | Delivery | ~24 months postpartum | 217 | 16 | 5 |
| **Objectives did not include the identification of predictors of maternal HIV acquisition during pregnancy and/or breastfeeding** | | | | | | | | |
| Chetty (2017) | South Africa | 2010–2015 | Prospective cohort study nested within a surveillance system | ~LMP | ~8 weeks postpartum | 1,621 | 66 | 1 |
| Gray (2005) | Uganda | 1994–1999 | Prospective cohort study | Pregnancy (first HIV test) | ~12 months postpartum | 4,371 | 63 | 6 |
| Humphrey (2006) | Zimbabwe | 1997–2001 | Prospective cohort study | Delivery | ~12 months postpartum | 9,562 | 269 | 16 |
| Le Roux (2019) | South Africa | 2014–2017 | Prospective cohort study | Pregnancy *Information on gestational age at enrollment was not provided.* | ~12 months postpartum | 493 | 8 | 2 |
| Leroy (1994) [b] | Rwanda | 1988–1992 | Prospective cohort study | Delivery | ~36 months postpartum | 216 | 20 | 2 |
| Mayaphi (2018) | South Africa | 2012–2016 | Cross-sectional (NAAT) study | ~LMP | Pregnancy (first HIV test) | 7,886 | 48 | 10 |
| Ortblad (2022) | Botswana | 2018–2019 | Prospective cohort study nested within a surveillance system | Pregnancy (first HIV test) | Pregnancy (second HIV test) *The median time to HIV retesting during pregnancy was 92 days (IQR: 72 to 112 days)* | 7,940 | 17 | 3 |
| Rice (2020) | Kenya | 2018 | Cross-sectional (RITA) study | ~LMP *Depending on gestational age at enrollment, the risk period may include time pre-conception. The RITA used in this study had a mean duration of recency of 141 days.* | Pregnancy (first HIV test) | 2,364 | 10 | 5 |
| Woldesenbet (2020) | South Africa | 2017 | Cross-sectional (RITA) study | ~LMP *Depending on gestational age at enrollment, the risk period may include time pre-conception. The RITA used in this study had a mean duration of recency of 141 days.* | Pregnancy (first HIV test) | 22,530 | 136 | 7 |

ANC: Antenatal care; IQR: Interquartile range; LMP: Last menstrual period; NAAT: HIV RNA nucleic acid amplification test; RITA: Recent Infection Testing Algorithm

[a] Secondary analyses of same cohort evaluated by Humphrey (2006)

[b] Secondary analysis of same cohort evaluated by Van de Perre (1991)

**Table 2. QUIPS risk of bias assessment for included publications.**

| Author (Year) | Study Participation — The study sample represents the population of interest on key characteristics, sufficient to limit potential bias of the observed relationship between predictors and outcome. | Study Attrition — Loss to follow-up is not associated with key characteristics, sufficient to limit potential bias to the observed relationship between predictors and outcome. | Predictor Measurement — Predictors are adequately measured in study participants, sufficient to limit potential bias. | Outcome Measurement — Outcome is adequately measured in study participants, sufficient to limit potential bias. | Analysis and Reporting — The statistical analysis is appropriate for the design of the study and reporting is complete, limiting potential for presentation of invalid or spurious results. |
|---|---|---|---|---|---|
| **Objectives included the identification of predictors of maternal HIV acquisition during pregnancy and/or breastfeeding** | | | | | |
| Bulterys (1994) | Red | Grey | Green | Green | Yellow |
| De Schacht (2014a) | Red | Yellow | Yellow | Green | Yellow |
| De Schacht (2014b) | Red | Red | Yellow | Green | Yellow |
| Dinh (2015) | Green | Grey | Yellow | Yellow | Yellow |
| Egbe (2016) | Red | Grey | Yellow | Red | Yellow |
| Fatti (2016) | Green | Yellow | Green | Green | Yellow |
| Hira (1990) | Red | Red | Yellow | Green | Yellow |
| Kinuthia (2010) | Red | Grey | Green | Green | Yellow |
| Kinuthia (2015) | Red | Green | Green | Green | Yellow |
| Machekano (2018) | Red | Green | Green | Green | Green |
| Mbena (2014) | Green | Green | Green | Green | Green |
| Miotti (1994) | Red | Red | Green | Yellow | Green |
| Mussa (2023) | Green | Green | Green | Green | Green |
| Schumann (2020) | Red | Grey | Yellow | Green | Yellow |
| Taha (1998) | Red | Red | Green | Green | Yellow |
| Tavengwa (2007) | Red | Red | Green | Green | Yellow |
| Van de Perre (1991) | Red | Green | Green | Green | Yellow |
| **Objectives did not include the identification of predictors of maternal HIV acquisition during pregnancy and/or breastfeeding** | | | | | |
| Chetty (2017) | Green | Yellow | Green | Red | Yellow |
| Gray (2005) | Green | Yellow | Green | Green | Green |
| Humphrey (2006) | Red | Red | Green | Green | Yellow |
| Le Roux (2019) | Red | Green | Green | Yellow | Yellow |
| Leroy (1994) | Red | Green | Green | Green | Green |
| Mayaphi (2018) | Red | Grey | Yellow | Yellow | Green |
| Ortblad (2022) | Green | Red | Green | Green | Green |
| Rice (2020) | Green | Grey | Green | Yellow | Green |
| Woldesenbet (2020) | Green | Grey | Green | Yellow | Green |

Red = high risk of bias; Yellow = moderate risk of bias; Green = low risk of bias; Grey = not assessed given study design

because no publication specified, *a priori*, the list of predictors under evaluation, all publications were classified as having a moderate risk of reporting biases.

Publications used univariable regression models, chi-square tests, t-tests, and Wilcoxon rank-sum tests to assess the associations between 66 different predictors and maternal HIV acquisition during pregnancy and/or lactation. Predictors fell into six domains (Table 3): maternal sociodemographic characteristics (n = 12), sexual and reproductive health history (n = 7), characteristics of the index pregnancy (n = 15), physical and mental health (n = 6), behaviors and knowledge (n = 13), and partner characteristics (n = 13).

We excluded 47 predictors from our narrative synthesis because they were evaluated in fewer than four study populations. We also excluded gestational age at initial HIV test, since this variable directly corresponds with the duration of HIV risk, and geographic location and partner occupation, since there was no overlap in how these predictors were defined between studies. These exclusions resulted in a total of 16 predictors for our narrative synthesis. Included predictors were heterogeneously defined (S2 Table). Estimated associations were imprecise, as illustrated by median CLRs that were greater than 1. We summarize associations and trends in Table 4. Abstracted data are available in S3 Table.

## Maternal sociodemographics

**Age.**   Twenty publications estimated associations between maternal age and maternal HIV acquisition [23–42]. Most (15 of 20) reported inverse associations. Because there was insufficient evidence to support a clear dose-response relationship, we collapsed across categories to generate two common contrasts: <20 years vs. ≥20 years [23, 24, 34–36, 38], and <25 years vs. ≥25 years [24, 26, 28, 29, 32, 33, 35–39, 41, 42]. These analyses generally suggested that younger age was associated with increased HIV incidence (Fig 2). The median risk ratio comparing women <20 years to women ≥20 years was 1.5 (range: 0.8–5.1), while the median rate ratio was 1.3 (range: 1.2–1.4). Estimates were more varied when comparing women <25 years to women ≥25 years (median risk ratio: 1.2, range: 0.4–2.3; median rate ratio: 2.4, range: 1.2–3.9). Analyses using continuous variables yielded estimates that ranged from 0.90 (i.e. a 10% relative reduction in incidence per additional year of age) to 1.0 (i.e. a 0% relative reduction in incidence per additional year of age) [24, 25, 27, 31, 32, 35].

**Marital status.**   Eighteen publications estimated associations between marital status and maternal HIV acquisition [23–25, 27–29, 31, 34, 35, 37–40, 42–46]. When publications distinguished between types of unmarried women (e.g. women who were single, divorced, widowed, or in a relationship but not married), and compared HIV incidence within these subgroups to incidence among married women [23, 38, 39, 42], or among women who were married or cohabiting [24, 25, 28, 31, 43], there were no apparent differences in the magnitude or direction of associations across categories. We therefore collapsed across types of unmarried women to generate two common contrasts: not married vs. married [23, 27, 29, 34, 37–40, 42, 44], and not married or cohabiting vs. married or cohabiting [24, 25, 28, 31, 35, 39, 43, 45, 46]. Estimates generally suggested higher HIV incidence among unmarried women than among married women (median risk ratio: 2.3, range: 0.3–8.0; median rate ratio: 2.3, range: 1.2–3.5), and among women who were not married or cohabitating relative to those who were (median risk ratio: 1.7, range: 0.6–2.1; median rate ratio: 5.1, range: 2.1–8.9; Fig 3). One study that captured longitudinal data on marital status reported that changes to marital status were associated with HIV incidence [35]. In this study, incident marriage during the risk period was associated with decreased HIV acquisition risk while incident divorce or widowing was associated with increased HIV acquisition risk.

**Table 3. Predictors evaluated by included publications.**

| | Buterys (1994) | De Schacht (2014a) | De Schacht (2014b) | Dinh (2015) | Egbe (2016) | Fatti (2016) | Hira (1990) | Kinuthia (2010) | Kinuthia (2015) | Machekano (2018) | Mbena (2014) | Miotti (1994) | Mussa (2023) | Schumann (2019) | Taha (1998) | Tavengwa (2007) | Van de Perre (1991) | Chetty (2017) | Gray (2005) | Humphrey (2006) | Le Roux (2019) | Leroy (1994) | Mayaphi (2018) | Ortblad (2022) | Rice (2020) | Woldesenbet (2020) | Number of publications contributing estimate of associations |
|---|---|---|---|---|---|---|---|---|---|---|---|---|---|---|---|---|---|---|---|---|---|---|---|---|---|---|---|
| **Sociodemographic** | | | | | | | | | | | | | | | | | | | | | | | | | | | |
| Age | X | X | X | NR | X | X | NR | NR | X | X | X | X | X | X | X | DR | | X | X | X | | X | X | X | X | X | 20 |
| Marital status | X | X | X | X | X | | | X | X | X | X | X | X | X | | DR | X | | X | X | X | X | X | | X | X | 18 |
| Educational attainment | NR | X | X | X | X | | | X | X | X | | | X | X | | | | X | X | | | | | | | X | 11 |
| Socioeconomic status | NR | X | X | X | X | | | X | X | | | | | X | X | | | | | X | | | | | | | 8 |
| Polygamy | | X | X | | | | | X | X | | X | | | | | | | | | | | | | | | | 5 |
| Geographic location[a] | | X | X | | | | | X | | | | | X | | | | | | | | | | | | | X | 5 |
| Urbanicity | X | | | | | | | | | | X | | X | X | | | | | | | | | | X | | | 5 |
| Occupation | | | | | | | | | | | | | X | X | | | | | | X | | | | | | | 3 |
| Religion | | | | | | | | | | | | | X | X | | | | | | X | | | | | | | 2 |
| Nationality | | | | | | | | | | | | | X | | | | | | | | | | | X | | | 2 |
| Duration of marriage | | | | | | | | X | | | | | | | | | | | | | | | | | | | 1 |
| Travel | NR | | | | | | | | | | | | | | | | | | | | | | NR | | | | 0 |
| **Sexual and reproductive health history** | | | | | | | | | | | | | | | | | | | | | | | | | | | |
| Sexually transmitted infections | X | | NR | | X | | X | | X | X | X | X | X | X | X | | X | | X | X | | | X | | | | 11 |
| Reproductive history | X | X | X | NR | NR | | | X | NR | | | X | X | X | X | DR | | | | X | | | | | X | X | 10 |
| Age at coital debut | X | X | X | | | | | | X | | | | | | | | | | | | | | | | | | 4 |
| Use of hormonal contraceptives | X | | | | | | | | | | | X | | | | | | | | NR | | | | | | | 2 |
| Cervical ectopy | | | | | | | | | | | | X | | | | | | | | | | | | | | | 1 |
| Age at menarche | | | | | | | | | X | | | | | | | | | | | | | | | | | | 1 |
| Years since first sex | X | | | | | | | | | | | | | | | | | | | | | | | | | | 1 |
| **Characteristics of the index pregnancy** | | | | | | | | | | | | | | | | | | | | | | | | | | | |
| Gestational age at initial HIV test[a] | | X | | X | X | X | | | X | X | | | | | | | | | | | | | | | X | X | 8 |
| Interval between HIV tests | | | | | X | | | | | | | | | | | | | | | | | | | | | | 3 |
| Location of ANC | | | | | | | | | | | X | | | X | | | | | | | | | | | X | | 3 |
| Pregnancy interval | X | | | | | | | | | | | X | | | | | | | | | | | | | | | 2 |
| Pregnancy intention | | | | | | | | | | | | | | | | | | | | | X | | | | | | 1 |
| Tested for syphilis | | | | X | | | | | | | | | | | | | | | | | | | | | | | 1 |
| Facility-based delivery | | | | X | | | | | | | | | | | | | | | | | | | | | | | 1 |
| Physician-attended delivery | | | | X | | | | | | | | | | | | | | | | | | | | | | | 1 |
| Caesarean delivery | | | | X | | | | | | | | | | | | | | | | | | | | | | | 1 |
| Received contraceptive counseling following delivery | | | | | | | | | | | | | | | | X | | | | | | | | | | | |
| Practiced safer breastfeeding | | | | | | | | | | | | | | | | | | | | NR | | | | | | | 1 |
| Number of ANC visits | | | | NR | | | | | | | | | | | | | | | | | | | | | | | 0 |
| Gestational age at delivery | | | | NR | | | | | | | | | | | | | | | | | | | | | | | 0 |
| Birth weight of infant | | | | NR | | | | | | | | | | | | | | | | | | | | | | | 0 |
| Vital status of infant | | | | | | | | | | | | | | | | | | | | NR | | | | | | | 0 |
| **Physical and mental health** | | | | | | | | | | | | | | | | | | | | | | | | | | | |
| Mental health conditions | | | | | | | | | | | | | | X | | | | | | | | | | | | | 1 |
| Maternal mid-upper arm circumference | | | | | | | | | | | | | | | | | | | | X | | | | | | | 1 |

*(Continued)*

**Table 3.** (Continued)

| | Butterys (1994) | De Schacht (2014a) | De Schacht (2014b) | Dinh (2015) | Egbe (2016) | Fatti (2016) | Hira (1990) | Kimuthia (2010) | Kimuthia (2015) | Machekano (2018) | Mbena (2014) | Miotti (1994) | Mussa (2023) | Schumann (2019) | Taba (1998) | Tavengwa (2007) | Van de Perre (1991) | Chetty (2017) | Gray (2005) | Humphrey (2006) | Le Roux (2019) | Leroy (1994) | Mayaphi (2018) | Orthlad (2022) | Rice (2020) | Woldesenbet (2020) | Number of publications contributing estimate of associations |
|---|---|---|---|---|---|---|---|---|---|---|---|---|---|---|---|---|---|---|---|---|---|---|---|---|---|---|---|
| Maternal hemoglobin levels | | | | | | | | | | | | | | | | | | | | X | | | | | | | 1 |
| Maternal serum retinol levels | | | | | | | | | | | | | | | | | | | | X | | | | | | | 1 |
| Maternal supplementation with Vitamin A | | | | | | | | | | | | | | | | | | | | X | | | | | | | 1 |
| Ever tested for tuberculosis | | | | X | | | | | | | | | | | | | | | | | | | | | | | 1 |
| **Behaviors and knowledge** | | | | | | | | | | | | | | | | | | | | | | | | | | | |
| Multiple sex partners | X | X | X | | NR | | NR | | X | X | | NR | | X | X | | NR | | X | X | | | NR | | | | 9 |
| Condom use | | X | X | | | | | | | X | X | X | | X | | | NR | | X | | | | X | | | | 7 |
| Abstinence | | | | | | | | | X | X | | X | | X | | | | | X | X | | | | | | | 5 |
| HIV risk perception | | X | X | | | | | X | | | | X | | X | | X | | | | | | | | | | | 4 |
| Intravaginal drying | | X | X | | | | X | | X | | | X | | | | | | | | | | | | | | | 4 |
| Knowledge about PMTCT | | X | X | | X | | | | | | | | | | | | | | | | | | | | | | 3 |
| Intimate partner violence | | | X | | | NR | | X | | | | | | X | | | | | | | | | NR | | | | 3 |
| History of testing for HIV | | X | X | | X | | | | | | | | | | | | | | | | | | | | | | 3 |
| History of iatrogenic needle exposure or blood transfusions | | | | | | | X | | | | | | | | | | X | | | | | NR | | | | | 3 |
| History of substance use | | | | | | | | | | | | | | X | | | | | | | | | X | | | | 2 |
| History of sex work | | | | | | | | | X | | | | | X | | | | | | | | | | | | | 2 |
| History of sex under the influence of substances | | | | | | | | | | | | | | X | | | | | | | | | | | | | 1 |
| Intravaginal washing | | | | | | | | | X | | | | | | | | | | | | | | | | | | 1 |
| **Partner characteristics** | | | | | | | | | | | | | | | | | | | | | | | | | | | |
| HIV status | | X | | X | X | X | X | | X | X | X | | | X | | | | | | | | | X | | | | 10 |
| Travel | X | X | X | | | | | | | | X | | | X | | | | | | | | | | | | | 5 |
| Occupation [a] | | X | X | | | | | | | | | | | X | | | | | | X | | | | | | | 4 |
| Educational attainment | | X | X | | | | | | | | | | | X | | | | | | | | | | | | | 3 |
| Circumcision status | | | | | X | X | | | X | | | | | X | | | | | | | | | | | | | 3 |
| Perception that partner has multiple partners | | X | X | | | | | | | | | | | X | | | | | | | | | | | | | 3 |
| Age difference | | | | | | | | | X | | | | | X | | | | | | | | | | | | | 2 |
| History of talking about HIV or HIV prevention with partner | | | | | | | | | | | | | | X | | X | | | | | | | | | | | 2 |
| History of HIV testing | | | X | | | | | | X | | | | | | | | | | | | | | | | | | 2 |
| History of couple HIV testing | | | | | | | | | | | | | | X | | | | | | | | | | | | X | 2 |
| Age | | | | | | | | | | | | | | X | | | | | | | | | | | | | 1 |
| Ever used substances | | | | | | | | | | | | | | X | | | | | | | | | | | | | 1 |
| Partner attended ANC visits | | | | | | | | | | | | | | X | | | | | | | | | | | | | 1 |

ANC: Antenatal care; DR: Evaluated as a predictor but not included in narrative review due to duplicity with results from primary study publication; NR: Predictor was measured by publication but no estimate of association was reported and tabulate data are unavailable; PMTCT: Prevention of mother-to-child transmission.

[a] Excluded from synthesis

**Table 4. Summary of associations between predictors of interest and risk of maternal HIV acquisition.**

| Predictor | Overall trends | Common contrast [a] | Summary of magnitude and precision [b] | | | | |
|---|---|---|---|---|---|---|---|
| | | | Risk Ratio | | Rate Ratio | | Median Confidence Limit Ratio (range) [c] |
| | | | Number of estimates | Median (range) | Number of estimates | Median (range) | |
| **Sociodemographic characteristics** | | | | | | | |
| Age | • 15 of 20 publications reported an inverse association between age and maternal HIV acquisition.<br>• 2 of 20 publications reported a positive association between age and maternal HIV acquisition.<br>• 2 of 20 publications reported a non-linear association.<br>• 1 of 20 publications reported that the direction of association was modified by reproductive status (pregnancy or lactation). | < 20 years of age vs. ≥ 20 years of age | 3 | 1.5 (0.8, 5.1) | 3 | 1.3 (1.1, 1.5) | 4.2 (1.7, 22.3) |
| | | < 25 years of age vs. ≥ 25 years of age | 7 | 1.2 (0.4, 2.3) | 5 | 2.4 (1.2, 3.9) | 5.3 (1.7, 12.6) |
| | | Continuous | 2 | 0.94 (0.9, 1.0) | 5 | 1.0 (0.9, 1.0) | 1.1 (1.1, 1.3) |
| Marital status | • 14 of 18 publications reported HIV incidence was lowest among women who were married or married and/or cohabiting.<br>• 4 of 18 publications reported HIV incidence was lowest among women who were single, separated, or in a relationship but not married. | Not married vs. Married | 8 | 2.3 (0.3, 8.0) | 2 | 2.3 (1.2, 3.5) | 6.4 (3.1, 17.4) |
| | | Not married/ cohabiting vs. Married or cohabiting | 6 | 1.7 (0.6, 11.7) | 3 | 5.1 (2.1, 8.9) | 5.3 (1.8, 377.2) |
| Educational attainment | • 6 of 11 publications reported an inverse association between educational attainment and maternal HIV acquisition.<br>• 2 of 11 publications reported a positive association between educational attainment and maternal HIV acquisition.<br>• 3 of 11 publications reported an inverted u-shaped trend between educational attainment and maternal HIV acquisition. | Undefined | - | - | - | - | - |
| Socioeconomic status | • 4 of 4 publications reported history of employment—or current employment—increased risk of maternal HIV acquisition.<br>• 4 of 4 publications reported risk of maternal HIV acquisition was higher among women with fewer assets compared to women with more assets.<br>• 1 of 1 publication reported a non-linear trend between household income and risk of maternal HIV acquisition. | Undefined | - | - | - | - | - |
| Polygyny | • 5 of 5 publications reported polygamous marriages increased risk of maternal HIV acquisition relative to monogamous marriages. | Polygamous marriage vs. Monogamous marriage | 2 | 5.9 (3.2, 8.7) | 3 | 1.8 (1.8, 2.4) | 6.3 (3.8, 24.5) |
| Urbanicity | • 2 of 5 publications reported that living in a rural setting increased risk of HIV acquisition relative to living in urban settings. The remaining three publications reported the opposite. | Rural vs. Urban residence | 4 | 0.8 (0.4, 1.8) | 1 | 2.5 | 5.6 (1.7, 17.1) |
| **Sexual and reproductive health** | | | | | | | |
| STIs | • 4 of 4 publications reported that a laboratory-confirmed diagnosis of any STI (*C. trachomatis*, *N. gonorrhoeae*, *T. vaginalis*, or *T. pallidum*) at the beginning of the risk period or at any point during the risk period increased HIV incidence during pregnancy and lactation.<br>• 3 of 4 publications reported that unspecified STIs increased HIV incidence during pregnancy and lactation. The remaining publication reported lower incidence among women who reported an STI in the previous three months than among women who reported no STI in the previous three months. This publication ascertained STI information at the time of testing for acute HIV infection during pregnancy.<br>• 5 of 6 publications reported that STI-related symptoms increased HIV incidence during pregnancy and lactation. The remaining publication suggested risk was higher among women who self-reported genital ulceration but not always among women who self-reported abnormal vaginal discharge. | STI vs. No STI *(Laboratory diagnosis)* | 8 | 3.3 (1.0, 62.7) | 4 | 3.8 (1.4, 9.2) | 14.6 (2.5, 54.1) |
| | | STI vs. No STI *(Self-reported, unspecified)* | 2 | 2.3 (0.7, 6.3) | 2 | 2.3 (1.2, 3.5) | 13.4 (4.5, 52.1) |
| | | STI-related symptoms vs. No STI-related symptoms *(Self-reported)* | 7 | 3.3 (0.6, 12.1) | 2 | 3.9 (2.1, 5.6) | 98.3 (4.1, 242.0) |
| Reproductive history | • 4 of 6 publications reported that HIV incidence was higher among women with fewer prior pregnancies than among women with more. The remaining two publications reported the opposite.<br>• 3 of 4 publications reported that HIV incidence was higher among women with fewer prior deliveries compared to women with more deliveries. The remaining publication reported the opposite.<br>• 1 of 2 publications reported that HIV incidence was higher among women with more children compared to women with fewer. The remaining publication reported the opposite. | Fewer prior pregnancies vs. More prior pregnancies | 4 | 1.0 (0.8, 4.2) | 2 | 1.5 (1.4, 1.6) | 9.2 (1.7, 17.5) |
| | | < 2 prior deliveries vs. ≥ 2 prior deliveries | 2 | 1.9 (1.1, 2.8) | 2 | 1.1 (1.0, 1.2) | 3.1 (1.6, 8.1) |
| | | Fewer children vs. More children | 1 | 0.6 | 1 | 2.1 | 3.5 (3.4, 3.6) |

*(Continued)*

**Table 4.** (Continued)

| Predictor | Overall trends | Common contrast [a] | Summary of magnitude and precision [b] | | | | Median Confidence Limit Ratio (range) [c] |
|---|---|---|---|---|---|---|---|
| | | | Risk Ratio | | Rate Ratio | | |
| | | | Number of estimates | Median (range) | Number of estimates | Median (range) | |
| Age at coital debut | • 4 of 4 publications reported an inverse association between age at coital debut and risk of maternal HIV acquisition. | Younger age vs. Older age | 1 | 3.4 | 2 | 2.8 (1.8, 3.8) | 4.2 (4.1, 13.8) |
| | | Continuous | 0 | - | 2 | 0.9 (0.8, 1.0) | 1.4 (1.4, 1.5) |
| **Behaviors and knowledge** | | | | | | | |
| Multiple partners | • 9 of 9 publications reported multiple partnerships increased risk of maternal HIV acquisition. | ≥ 2 sex partners vs. < 2 sex partners | 3 | 1.9 (1.3, 10.1) | 7 | 3.9 (1.3, 22.3) | 8.7 (3.93, 57.92) |
| | | Continuous | 0 | - | 1 | 1.2 | 1.3 |
| Condom use | • 3 of 4 publications suggested risk of maternal HIV acquisition was higher among women who reported consistent condom use compared to women who reported inconsistent or no condom use. The remaining publications suggested the opposite. <br> • 4 of 6 publications suggested risk of maternal HIV acquisition was higher among women who reported any condom use compared to women who reported none. The remaining two publications suggested the opposite. | Consistent condom use vs. Inconsistent/no condom use | 2 | 6.0 (3.9, 8.1) | 2 | 0.8 (0.4, 1.2) | 48.6 (7.0, 52.0) |
| | | Any condom use vs. No condom use | 4 | 1.6 (0.7, 5.7) | 2 | 1.5 (1.0, 2.0) | 8.3 (3.2, 19.6) |
| Abstinence | • 3 of 5 publications suggested risk of maternal HIV acquisition was lower among women who reported any abstinence compared to women who reported no abstinence. The remaining two publications suggested the opposite. | Abstinent vs. Not abstinent | 2 | 0.7 (0.6, 0.8) | 2 | 1.1 (0.6, 1.6) | 12.6 (5.7, 17.0) |
| | | Continuous (in months) | 0 | - | 1 | 4.1 | 6.5 |
| HIV risk perception | • 4 of 4 publications reported perceiving oneself to be vulnerable to HIV increased risk of maternal HIV acquisition. | Felt vulnerable to HIV vs. Did not feel vulnerable to HIV | 3 | 2.0 (1.0, 2.5) | 1 | 4.2 | 11.7 (3.2, 51.9) |
| Intravaginal drying | • 4 of 4 publications reported a positive association between history of intravaginal drying and risk of maternal HIV acquisition. | Any vaginal drying vs. No vaginal drying | 3 | 1.8 (1.6, 26.9) | 1 | 1.1 | 6.2 (4.0, 14.2) |
| **Partner characteristics** | | | | | | | |
| HIV status | • 5 of 7 publications suggested risk of maternal HIV acquisition was higher among women who did not know their partner's HIV status compared to women who reported knowing their partner's HIV status. The remaining two publications reported the opposite. <br> • 3 of 3 publications suggested risk of maternal HIV acquisition was higher among women who reported partner previously diagnosed with HIV compared to women who reported an HIV-negative partner. <br> • 4 of 4 publications suggested risk of maternal HIV acquisition was higher among women who reported a partner previously diagnosed with HIV or a partner with an unknown HIV status compared to women who reported an HIV-negative partner. | Unaware of partner status vs. Aware of partner status | 6 | 1.5 (0.5, 3.1) | 1 | 2.4 | 4.7 (1.8, 12.2) |
| | | Partner living with HIV vs. Partner HIV-negative | 2 | 2.8 (1.6, 3.8) | 1 | 3.9 | 62.0 (50.6, 73.3) [d] |
| | | Partner living with HIV or unknown status vs. Partner HIV-negative | 3 | 1.6 (0.7, 2.5) | 1 | 2.5 | 4.5 (3.6, 7.9) |
| Partner travel | • 3 of 5 publications suggested a positive association between having a male partner who traveled frequently or for long periods of time and risk of maternal HIV acquisition. The remaining two publications reported the opposite. | Partner travels vs. Partner does not travel | 2 | 8.2 (7.3, 9.1) | 3 | 0.7 (0.6, 1.4) | 7.4 (4.7, 10.0) |

[a] Publications may contribute estimates to more than one contrast.

[b] When only one estimate is available, we present the estimate of association instead of the median and range of estimated associations.

[c] Because the CLR is a standardized metric of precision, the median and range offer a summary of precision of all estimated associations for a common contrast, regardless of measure. A ratio closer to 1 indicates a more precise estimate while a ratio farther from 1 indicates a less precise estimate.

[d] One publication only reported a point estimate and did not provide tabulated data to generate a confidence interval with. CLR is therefore based on incomplete information

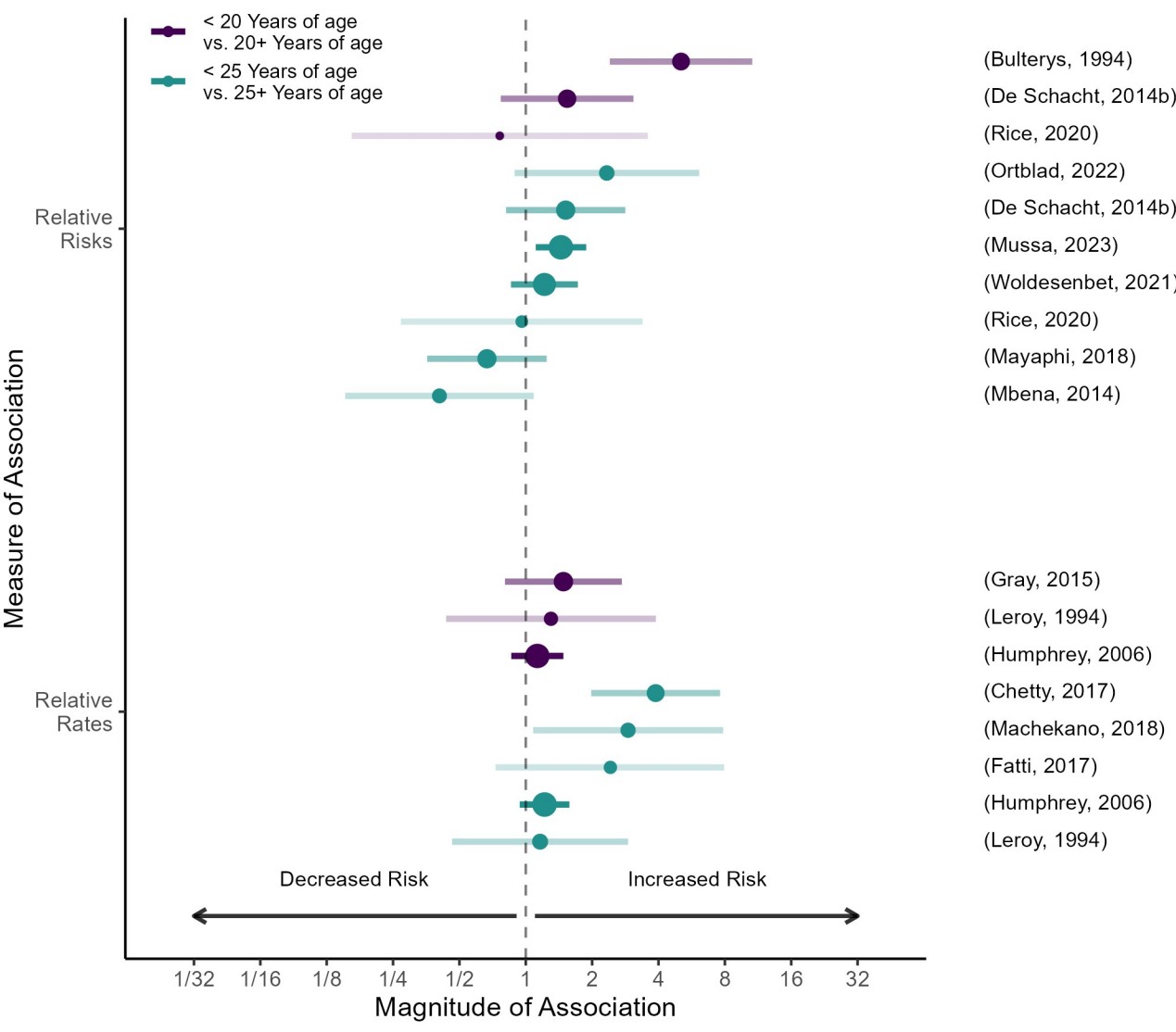

The size of the point estimate and the opacity of the confidence interval are inversely proportional to the confidence limit ratio.

**Fig 2. Associations between age and maternal HIV acquisition.**

**Educational attainment.** Eleven publications estimated associations between educational attainment and maternal HIV acquisition [24, 25, 27, 28, 31, 34, 39, 40, 42–44]. Estimates of association varied in direction: six publications reported inverse associations [24, 27, 31, 40, 42, 44], two reported positive associations [25, 34], and three reported non-linear trends [28, 39, 43]. Because several publications did not clearly define categories used in analyses, we were unable to generate standardized contrasts for comparison.

**Socioeconomic status (SES).** Eight publications estimated associations between SES and maternal HIV acquisition. SES was measured in terms of history of employment [24, 25, 40, 44], household income [35], single asset ownership [32, 44], or asset-based indices of household wealth [31, 43]. These measures had important limitations that precluded a summary of trends: employment was not defined, household income was measured as a categorical variable

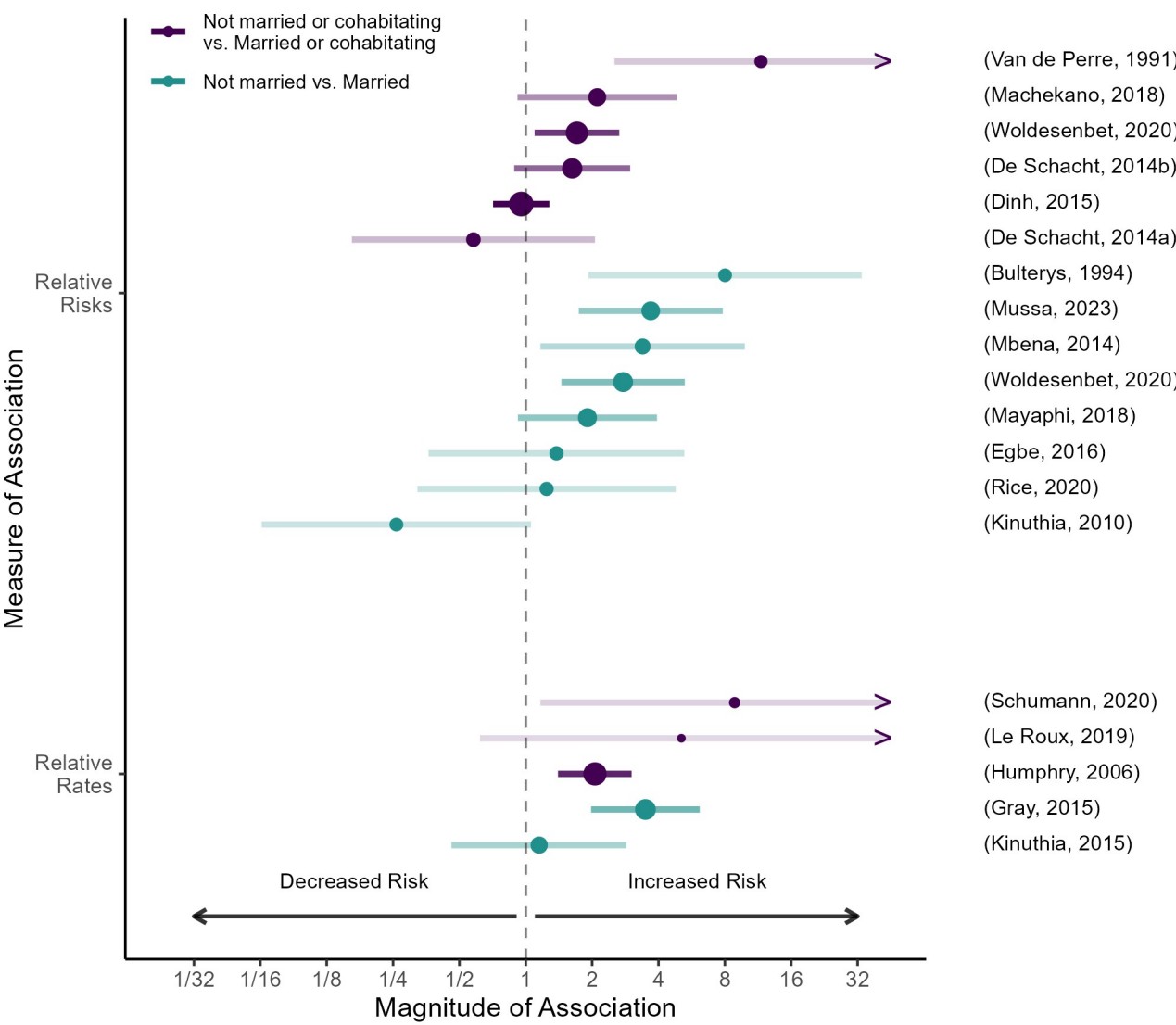

The size of the point estimate and the opacity of the confidence interval are inversely proportional to the confidence limit ratio.

**Fig 3. Associations between marital status and maternal HIV acquisition.**

with a category for missingness, and the sensitivity of asset-based SES measures to urban-rural differences were not accounted for in analyses.

**Polygyny.** Five publications estimated associations between polygyny and maternal HIV acquisition [24, 25, 27, 29, 44]. All five publications restricted their analyses to married women and compared HIV incidence among women in a polygynous marriage to incidence among women in a monogamous marriage. All estimates suggested that polygyny was associated with an increased HIV incidence (median risk ratio: 5.9, range: 3.2–8.7; median rate ratio: 1.8, range: 1.8–2.4).

**Urbanicity.** Five publications estimated associations between urbanicity and maternal HIV acquisition [23, 29, 31, 41, 42]. There was limited evidence that HIV risk differed according to urbanicity of residence.

## Sexual and reproductive health

**Sexually transmitted infections (STIs).** Eleven publications contributed 25 estimates of association between STIs and maternal HIV acquisition [23, 27–32, 34, 37, 45, 47]. Most estimates (23 of 25) suggested greater HIV incidence among PLW with STIs compared to PLW without (median risk ratio: 3.3; range: 0.6–62.7; median rate ratio was 3.2, range: 1.1–9.2; Fig 4). There were important variations in how publications measured and defined STIs.

Four publications measured unspecified STIs (self-reported) over different time periods: before the start of the risk period (i.e. before first negative HIV test) [27], during the risk period (i.e. between first negative HIV test and final HIV test) [23], and during pregnancy/in the last three months [31, 37], which included time before and during the risk period. Three of the four estimates suggested a positive association between unspecified STIs and maternal

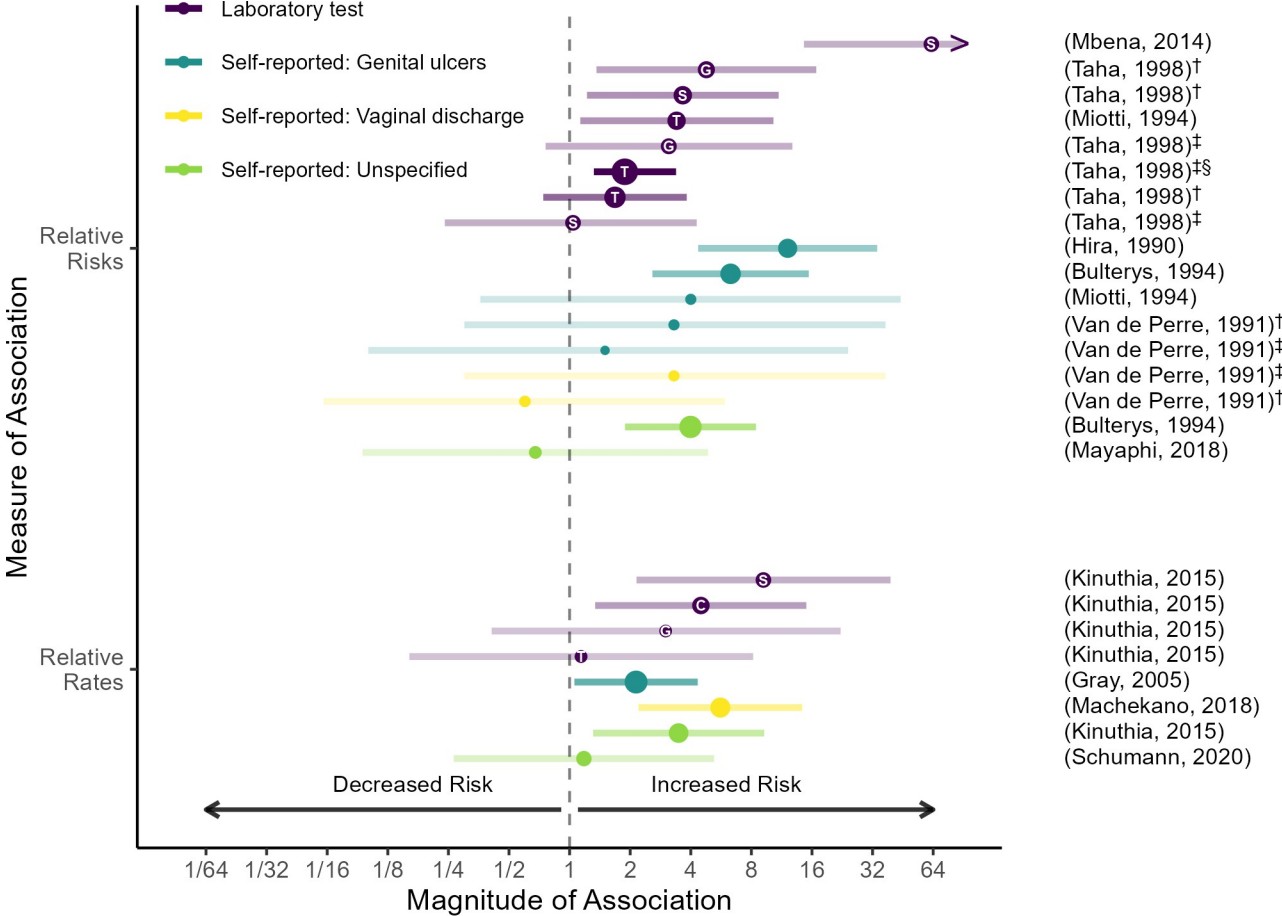

The size of the point estimate and the opacity of the confidence interval are inversely proportional to the confidence limit ratio. For laboratory tests, the specific STI diagnosed is denoted by the capital letter placed inside the point estimate: "S" for syphilis, "T" for trichomonas, "G" for gonorrhoea, and "C" for clamydia.

†Specific to HIV acquisition during pregnancy
‡Specific to HIV acquisition in the postpartum period
§Confidence interval is not symmetric around point estimate, reflecting data reported in the published text.

**Fig 4. Associations between sexually transmitted infections and maternal HIV acquisition.**

HIV acquisition (median risk ratio: 2.3, range: 0.7–4.0; median rate ratio: 2.3, range: 1.2–3.5). The publication that reported an inverse association asked pregnant women about STIs in the three months prior to their test for acute HIV infection [37].

Six publications measured STI-related symptoms (self-reported). Four defined STI-related symptoms as genital ulcerations (GUD) during the risk period [23, 30, 34, 47], one defined it as abnormal vaginal discharge (AVD) during the risk period [28], and one defined it in four ways (GUD before the risk period, GUD during the risk period, AVD before the risk period, and AVD during the risk period) [45]. Eight of nine estimates suggested a positive association between self-reported STI-related symptoms and maternal HIV acquisition (median risk ratio: 3.3, range: 0.6–12.1; median rate ratio: 3.9, range: 2.1–5.6). The publication that reported an inverse association asked pregnant women about AVD in the three months prior to the risk period [45].

Finally, four publications tested participants for syphilis [27, 29, 32], trichomoniasis [27, 30, 32], gonorrhea [27, 32], and/or chlamydia [27] at the beginning of the risk period (i.e. on the day of the initial HIV negative test). In addition, one publication routinely tested participants for syphilis, trichomoniasis, and gonorrhea at six-month intervals during the risk period [32]. All estimates suggested that a diagnosed STI was associated with increased maternal HIV acquisition (median risk ratio: 3.3, range: 1.0–62.7; median rate ratio: 3.8, range: 1.4–9.2). The magnitude of association was generally larger when STIs were diagnosed at the beginning of the risk period (median risk ratio: 4.2, range: 1.7–62.7; median rate ratio: 3.8, range: 1.1–9.2) than during the risk period (median risk ratio: 1.9, range: 1.0–3.1), and for syphilis, gonorrhea, and chlamydia infections than for trichomoniasis infections.

**Reproductive history.**   Ten publications estimated associations between prior pregnancies [23, 25, 31, 38, 39, 42], parity [30, 31, 35, 42], and number of living children [24, 44] and maternal HIV acquisition. There was some evidence that HIV risk was greater among women with fewer pregnancies or deliveries than among those with more.

**Age at coital debut.**   All four publications that estimated associations between age at coital debut and maternal HIV acquisition reported an inverse association, linking older age at debut with lower HIV incidence [23–25, 27]. Publications using continuous variables estimated associations ranging from 0.84 (i.e. a 16% relative reduction in risk per additional year of age at debut) to 0.98 (i.e. a 2% relative reduction in risk per additional year of age at debut) [24, 27]. Publications using categorical variables illustrated that the magnitude of association was sensitive to index and referent choices [23–25], with more extreme estimates when the age difference between index and reference group was larger.

## Behaviors and knowledge

**Multiple sex partners.**   All nine publications estimating associations between multiple sex partners and maternal HIV acquisition reported higher HIV incidence among PLW with more than one partner compared to those with only one partner (median risk ratio: 2.0; range: 1.3, 10.1; median rate ratio was 3.9, range: 1.3, 22.3; Fig 5). Notable variations existed in definitions across publications.

Four publications estimated associations between the number of sex partners before or at the start of the risk period the risk period and maternal HIV acquisition. Two of these publications compared HIV incidence among women who had ever had sex with someone other than their current partner to incidence among women who had only ever had sex with their current partner (median rate ratio: 10.4, range: 3.9–16.9) [24, 25]. One publication used a continuous linear variable representing lifetime sex partners and estimated a 20% relative increase in the incidence rate for each additional partner [27]. Finally, one publication reported that women

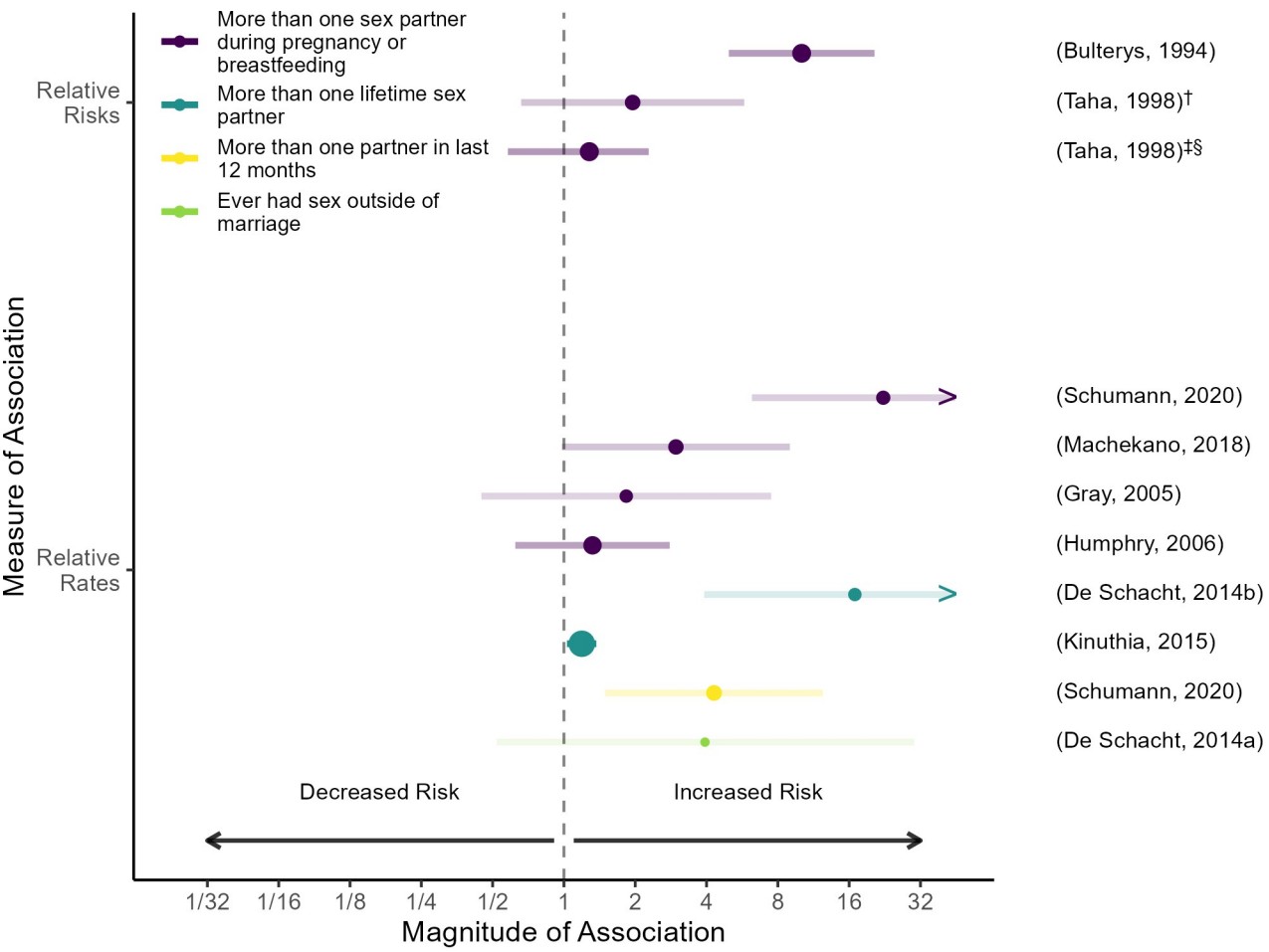

The size of the point estimate and the opacity of the confidence interval are inversely proportional to the confidence limit ratio.

†Specific to HIV acquisition during pregnancy
‡Specific to HIV acquisition in the postpartum period
§Confidence interval is not symmetric around point estimate, reflecting data reported in the published text.

**Fig 5. Associations between multiple sex partners and maternal HIV acquisition.**

with ≥2 partners at the beginning of the risk period had 1.9 times the risk of acquiring HIV during pregnancy, and 1.3 times the risk of acquiring HIV during lactation, compared to those with one partner [32].

Four publications estimated associations between the number of sex partners during the risk period and maternal HIV acquisition. Two of these publications compared HIV incidence among women reporting ≥2 partners during the risk period to incidence among women reporting ≤1 partners during the risk period (median rate ratio: 2.4, range: 1.8–3.0) [28, 34]. One publication compared HIV incidence between women who did and did not have sex with a partner other than the infant's father during the risk period, and found a tenfold increase in incidence among those who did [23]. The fourth publication reported that women with ≥1 new partner during the risk period were 1.3 times as likely to acquire HIV as those with no new partners [35].

Lastly, one publication measured the number of partners in the last year and during the index pregnancy [31]. Both measures included time before and during the risk period. Among women who reported >1 partner in the last year, the HIV incidence rate was 4.3 times the rate among women who reported only one sex partner. This association strengthened when comparing women with >1 partner during the index pregnancy to those with only one partner (rate ratio: 22.3).

**Condom use.** Seven publications estimated associations between condom use and maternal HIV acquisition. Three measured condom use during the risk period [28, 30, 34], one measured condom use during pregnancy, which included time before and during the risk period [31], and three did not specify the timing of condom use relative to the risk period [24, 25, 37]. In general, maternal HIV incidence was higher among women who reported using condoms. When publications distinguished between women reporting consistent and inconsistent condom use, and compared HIV incidence within these subgroups to incidence among women reporting no condom use, there were no apparent differences in the magnitude or direction of associations across categories [25, 34, 37]. We therefore generated two common contrasts: any condom use vs. no condom use [24, 25, 30, 31, 34, 37], and consistent condom use vs. inconsistent or no condom use [25, 28, 34, 37]. Estimates generally indicated higher maternal HIV incidence among women reporting any condom use than among women reporting no condom use (median risk ratio: 1.6, range: 0.7–5.7; median rate ratio: 1.5, range: 1.0–2.0), and among women reporting consistent use compared to those reporting inconsistent use (median risk ratio: 0.8, range: 0.4–1.2; median rate ratio: 6.0, range: 3.9–8.1; Fig 6).

**Abstinence.** Five publications estimated associations between sexual abstinence and maternal HIV acquisition. Abstinence was measured in the 30 days prior to the risk period [27], during the risk period [28, 30, 35], and/or during pregnancy, which included time before and during the risk period [28, 31]. Overall, there was inconsistent evidence of an association across measure types (median risk ratio: 0.7, range: 0.6–0.8; median rate ratio: 1.6, range: 0.6–4.1). The publication that used the most granular measure of abstinence—months postpartum—reported that each month or abstinence in the postpartum period increased the hazard of HIV acquisition by 4.1% [35].

**HIV risk perception.** Four publications estimated associations between perceptions of HIV vulnerability and maternal HIV acquisition [24, 31, 44, 48]. Only one publication defined when—relative to the risk period—their measure of HIV risk perception applied. In this study, participants were asked about their perceived vulnerability in the last 12 months, which included time before and during the risk period under evaluation [44]. Estimates from all publications suggested higher HIV incidence among PLW who perceived themselves to be vulnerable to HIV relative to those who did not. Publications that distinguished between low, medium, and high perceived vulnerability provide some indication that HIV incidence may increase with increasing vulnerability perception, though confidence intervals overlapped considerably [24, 31].

**Intravaginal drying.** Four publications estimated associations between vaginal drying and maternal HIV acquisition [24, 27, 30, 47]. All estimates suggested a positive association between vaginal drying and maternal HIV acquisition. The risk of HIV among women who reported dry sex during the risk period was nearly 27 times the risk among women who did not report dry sex during the risk period [47]. Association were weaker (range: 1.1–1.8) when measures of vaginal drying corresponded to any drying in the week before the risk period [27], vaginal drying at any point before the risk period [24], or ever treating vaginal discharge [30].

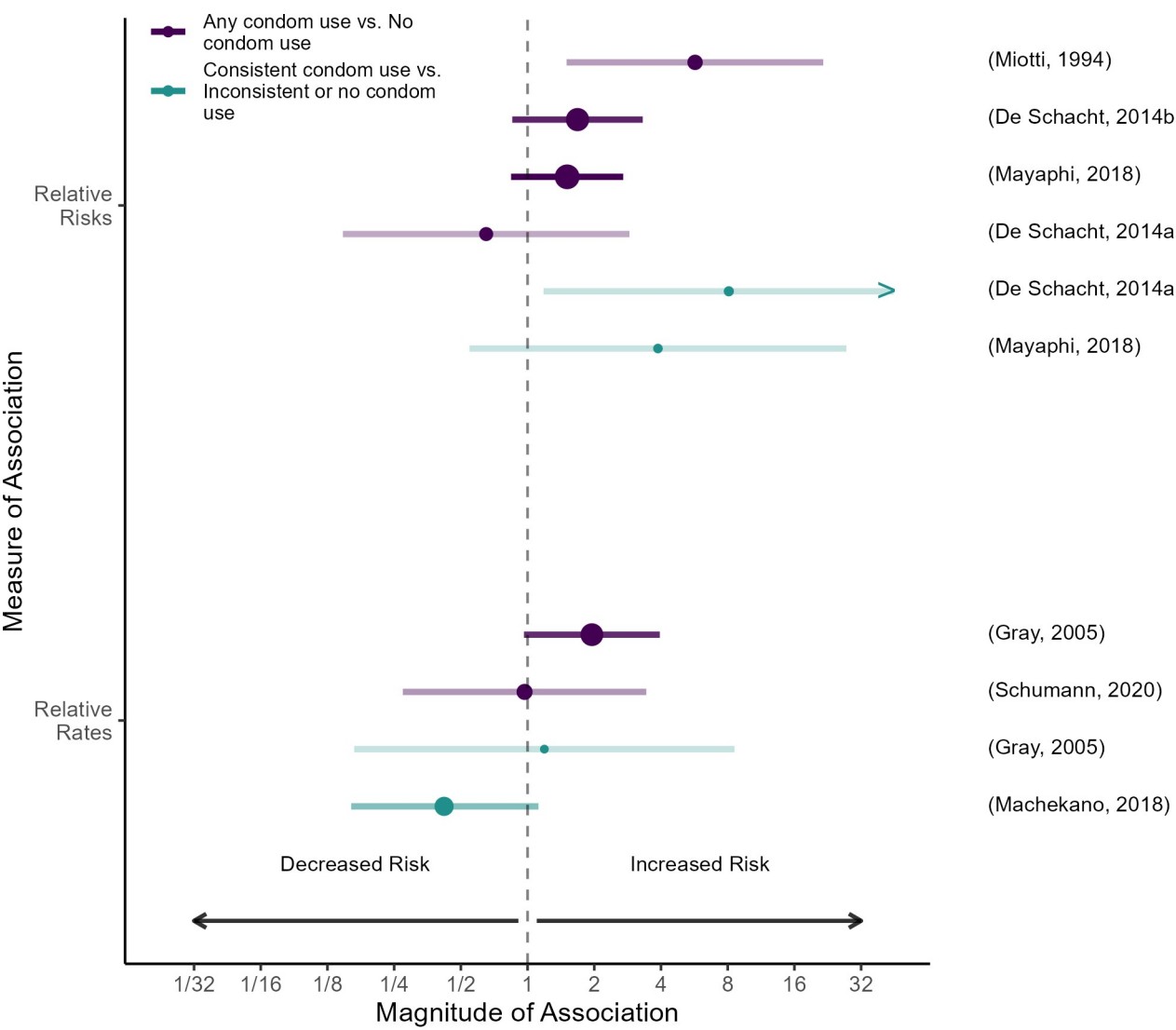

The size of the point estimate and the opacity of the confidence interval are inversely proportional to the confidence limit ratio.

**Fig 6. Associations between condom use and maternal HIV acquisition.**

## Partner characteristics

**HIV status.** Seven publications estimated associations between maternal awareness of partner HIV status and maternal HIV acquisition. These publications measured awareness at either the start of the risk period [25–28] or the end of the risk period (i.e. at the time of the final HIV test) [37, 40, 43]. Three of the four publications that measured awareness at the start of the risk period reported greater HIV incidence among women unaware of their partner's HIV status than women who were aware of the partner's HIV status (median relative risk: 2.7, range: 0.7–3.1). Two of the three publications that measured awareness at the end of the risk period also reported greater incidence among women who were unaware than those who were aware, though the association was somewhat weaker (median relative risk: 1.3, range: 0.5–1.6).

Six publications evaluated associations between couple HIV status and maternal HIV acquisition. In all publications, couple HIV status was determined based on maternal self-report. Two of these publications reported zero maternal seroconversions among women in HIV-serodifferent partnerships [27, 37], and one reported zero maternal seroconversions among women in HIV-negative concordant partnerships [26]. Estimates from the remaining three publications suggested that HIV incidence was greater among women in known serodifferent partnerships than among women in HIV-negative concordant partnerships (median risk ratio: 2.7, range: 1.6–3.8) [25, 28, 47]. Similar trends were observed when HIV incidence was compared between women who reported a partner living with HIV or a partner with an unknown HIV status and women who reported an HIV-negative partner (median risk ratio: 1.6, IQR: 0.7–2.5) [27, 28, 31, 37].

Notably, all publications asked PLW only about their primary partner's HIV status; no measure captured whether the woman was aware of all partner HIV statuses if she had more than one partner.

**Partner travel.**    Five publications estimated associations between partner travel and maternal HIV acquisition. While estimates from three publications suggested an increased HIV incidence among women whose partners traveled frequently or for extended periods of time compared to women whose partners did not travel [23, 29, 31], estimates from the other two publications suggested the opposite [24, 25].

## Discussion

In this systematic review, we identified 26 publications that evaluated a diverse range of sociodemographic, clinical, behavioral, and partner-level predictors of HIV acquisition among PLW in SSA. Estimates from individual publications were generally imprecise, and substantial variability in how publications measured and analyzed predictors precluded meta-analyses. Nevertheless, we identified trends between commonly measured characteristics and maternal HIV acquisition during pregnancy and/or lactation. Our findings illustrate that HIV risk among PLW is not uniform, contextualizing some nuance around the high average risk of maternal HIV infection estimated in previous work [5, 6].

Our study makes an important contribution to our understanding of HIV acquisition among PLW. Most studies included in this review contributed estimates of association that were imprecise and lacked generalizability. Such results are difficult to interpret in isolation. By summarizing results across individual studies, our review offers insights into the directional consistency of estimated associations across a wide range of populations studied over several decades. We observed greater HIV incidence among young PLW, a finding that reflects increased vulnerability to HIV among adolescent girls and young women in SSA [49–52]. In most studies, PLW in stable, monogamous, relationships were less likely to acquire HIV than those who were single, separated, or in a polygynous marriage. Our findings suggest that perceiving oneself to be at risk of HIV was associated with greater HIV risk during pregnancy and lactation, as was early coital debut and dry sex. Multiple partnerships, STIs, and partner HIV status were also associated with HIV incidence among PLW. Interestingly, although correct condom use is effective at preventing HIV acquisition [53], most studies found that condom use was associated with increased HIV acquisition among PLW. This unexpected association may be a function of reporting biases [54] or crude condom use measures. It may also reflect condom use as a marker for HIV exposure risk since, in SSA, condoms are typically used in the context of relationships perceived as carrying higher HIV acquisition risk [55, 56], and condom use often decreases during pregnancy and the postpartum period [8–10].

While trends based on imprecise estimates should be interpreted cautiously, consistency in the direction of association across time and place provides greater confidence in the relationship between a predictor and maternal HIV acquisition [12]. Of note, however, our findings are primarily derived from data collected in southern and eastern Africa prior to the widespread implementation of combination HIV services. Although plausible biological and behavioral links between many predictors identified in this review and HIV acquisition suggest their continued relevance in these regions as the epidemic landscape evolves, it remains uncertain whether our findings can be generalized to central and western Africa where the HIV epidemic is primarily concentrated among key populations. Understanding what factors predict HIV risk among PLW in central and western Africa is critical, particularly as the proportion of new pediatric HIV infections attributed to maternal HIV seroconversion is increasing [57].

Global institutions recommend using HIV risk assessment tools to guide conversations between providers and patients about potential HIV exposures and HIV prevention strategies, including PrEP [4, 11]. This guidance has been widely adopted across Africa, where policy documents structuring PrEP implementation often specify that PLW are eligible for PrEP if they are at "high risk" of acquiring HIV [58]. Validated HIV risk assessment tools [59], however, have demonstrated only moderate ability to predict HIV acquisition in external study populations [60], and results from a clinical trial conducted in Kenya suggested targeted PrEP offer was no more effective than universal PrEP offer in reducing HIV incidence among PLW [61]. Offering PrEP to all PLW and allowing women to make decisions according to their self-perceived risk may simplify PrEP delivery, decrease stigma related to PrEP use, and overcome challenges with measuring HIV risk. Notably, while HIV risk perception was consistently associated with HIV acquisition in this review, approximately 40% of PLW who seroconverted in contributing studies reported no or limited risk of HIV acquisition. These findings echo results from Girl Power-Malawi and FEM-PrEP, which also illustrated discrepancies between perceived and actual HIV risk [62, 63]. Developing better tools for providers and individuals to assess HIV risk, and evaluating how to implement such tools, is needed to optimize PrEP implementation in the region, particularly as more effective—and expensive—PrEP modalities become more widely available [64]. This review offers insights into candidate variables that may support the development and refinement of such tools.

Studies included in this review primarily evaluated individual-level predictors of HIV acquisition. HIV risk, however, is shaped by combinations of individual- and partner-level behaviors and characteristics, some of which may be reflected by population-level proxies in certain contexts. Age-discordant partnerships and the perception that the partner has multiple partners have been included in HIV risk assessment tools for women in SSA [65], and subnational prevalence of viremia (defined as HIV RNA ≥1000 copies/mL) was the single most important predictor of HIV risk for women aged 15–49 in a pooled analysis of Population HIV Impact Assessments from 15 African countries [66]. While studies included in this review reported positive associations between age-discordant partnerships and the perception that the partner had other partners and maternal HIV acquisition, no study considered whether population-level measures were associated with maternal HIV acquisition. Understanding how such factors might shape HIV risk in the context of pregnancy and lactation could help guide national decision-making around where to offer PrEP and align implementation strategies with global guidance [4, 11].

While many of the predictors identified in our review are known cofactors of HIV acquisition, our approach facilitated an examination of the nuances related predictor measurement during antenatal and postnatal care. For example, most attributes associated with maternal HIV acquisition in this literature are dynamic, meaning that they can evolve over time according to changes to internal and external circumstances. By assessing when, relative to the risk

period, predictors were measured, we were able to illustrate that specific events during pregnancy and lactation (e.g. changes to marital status, new STI diagnoses or symptoms, new sex partners, and new information about partner HIV status) were frequently associated with HIV acquisition. Such findings are consistent with the "seasons of risk" literature, which highlights how risk is often episodic and may change rapidly over short intervals [67]. However, because of temporal and definitional variations across studies in terms of risk periods, predictor and outcome assessments, and their interrelationships, additional research is needed to improve our understanding of HIV risk trajectories over pregnancy and lactation to inform dynamic HIV prevention approaches for antenatal and postnatal settings in SSA.

Because our narrative synthesis focused on unadjusted estimates of association—a decision we made to facilitate comparison across studies [14]—we caution against considering predictors as targets for intervention [68]. Unadjusted associations may not correspond to causal effects, and intervening on non-causal predictors may be ineffective or even have unintended, adverse consequences. For example, while our results suggest an association between condom use and maternal HIV acquisition in SSA, using condoms does not cause HIV acquisition, and efforts to reduce condom use during pregnancy and lactation would likely increase HIV acquisition risk during these periods. To determine whether modifiable predictors of maternal HIV acquisition identified in this review should be targets for intervention, a better understanding of the underlying causal relationships would be required [68].

Our results should be interpreted considering the following limitations. First, because our approach did not account for differences in the relative size of studies, imprecise estimates were given the same weight as more precise estimates in determining overall trends. We note, however, that estimates from larger studies typically suggested the same direction of association as estimates from smaller studies. Second, we purposefully excluded conference abstracts on the basis that they were unlikely to provide rich detail on measurement necessary for narrative synthesis and were more likely to emphasize non-null results than peer-reviewed manuscripts [69]. This decision may have excluded relevant studies from our narrative synthesis. Third, among publications included in this review, none specified how candidate predictors were selected for analysis, and 13 of the 26 publications did not provide an estimate of association for each of the predictors that they measured. While these features may introduce selective reporting bias, no publication included in this review reported only statistically significant (i.e. $p < 0.05$) findings, and two publications reported only statistically non-significant results (i.e. $p \geq 0.05$), indicating that studies with null results are contributing to this review. Finally, for predictors where there was no strong evidence that the direction or magnitude of association differed according to how the predictor variable was defined, we generated common contrasts and estimated associations using reported data. While this approach allowed us to provide insights across studies, dichotomizing variables may obscure more nuanced relationships between predictors and maternal HIV acquisition risk.

## Conclusions

To achieve the ambitious goal of eliminating vertical HIV transmission, PMTCT programs in SSA must provide interventions to prevent maternal HIV acquisition during pregnancy and lactation. The WHO and UNAIDS recommend a risk-guided approach to implementing HIV prevention interventions in high-burden, low-resource, antenatal and postnatal settings. This review offers insights on candidate attributes that are feasible to measure and broadly applicable to PLW in SSA. These data may contribute to the development and refinement of risk assessment tools to identify those PLW who are most likely to benefit from access to comprehensive HIV prevention.

## Supporting information

**S1 Table. Database search strategy.**
(DOCX)

**S2 Table. Summary of predictor operationalization in each publication.**
(DOCX)

**S3 Table. Extraction file.**
(XLSX)

## Author Contributions

**Conceptualization:** Lauren A. Graybill, Benjamin H. Chi, Nora E. Rosenberg, Kimberly A. Powers, Wilbroad Mutale.

**Data curation:** Lauren A. Graybill, Jennifer S. Bissram.

**Formal analysis:** Lauren A. Graybill, Benjamin H. Chi, Twaambo E. Hamoonga, Jasmine N. Hodges, Brian D. Richardson, Kellie Freeborn.

**Funding acquisition:** Benjamin H. Chi, Wilbroad Mutale.

**Supervision:** Benjamin H. Chi.

**Writing – original draft:** Lauren A. Graybill.

**Writing – review & editing:** Benjamin H. Chi, Twaambo E. Hamoonga, Margaret Kasaro, Jasmine N. Hodges, Brian D. Richardson, Jennifer S. Bissram, Friday Saidi, Katie R. Mollan, Kellie Freeborn, Nora E. Rosenberg, Kimberly A. Powers, Wilbroad Mutale.

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
