## [Decision Letter · Decision Letter 0]

9 Sep 2024

PONE-D-24-20929Predictors of maternal HIV acquisition during pregnancy and lactation in sub-Saharan Africa:  a systematic review and narrative synthesisPLOS ONE

Dear Dr. Graybill,

Thank you for submitting your manuscript to PLOS ONE. After careful consideration, we feel that it has merit but does not fully meet PLOS ONE’s publication criteria as it currently stands. Therefore, we invite you to submit a revised version of the manuscript that addresses the points raised during the review process.

We look forward to receiving your revised manuscript.

Kind regards,

Rabie Adel El Arab

Academic Editor

PLOS ONE

Journal requirements: 1. When submitting your revision, we need you to address these additional requirements. Please ensure that your manuscript meets PLOS ONE's style requirements, including those for file naming. The PLOS ONE style templates can be found at https://journals.plos.org/plosone/s/file?id=wjVg/PLOSOne_formatting_sample_main_body.pdf and https://journals.plos.org/plosone/s/file?id=ba62/PLOSOne_formatting_sample_title_authors_affiliations.pdf. 2. Thank you for stating the following financial disclosure:  [This study was funded by the National Institute of Allergy and Infectious Diseases (NIAID) through award R01 AI131060. Additional investigator, trainee, and administrative support was provided by NIAID (T32 AI007001, K24 AI120796, P30 AI050410, R01 AI157859), and the Fogarty International Center (D43 TW009340, D43 TW010558).].  Please state what role the funders took in the study.  If the funders had no role, please state: ""The funders had no role in study design, data collection and analysis, decision to publish, or preparation of the manuscript."" If this statement is not correct you must amend it as needed. Please include this amended Role of Funder statement in your cover letter; we will change the online submission form on your behalf. 3. Thank you for stating the following in the Competing Interests section: [Drs. Graybill and Chi received consulting fees from UNICEF. All other authors reported no conflicts of interest.].  Please confirm that this does not alter your adherence to all PLOS ONE policies on sharing data and materials, by including the following statement: ""This does not alter our adherence to  PLOS ONE policies on sharing data and materials.” (as detailed online in our guide for authors http://journals.plos.org/plosone/s/competing-interests).  If there are restrictions on sharing of data and/or materials, please state these. Please note that we cannot proceed with consideration of your article until this information has been declared.  Please include your updated Competing Interests statement in your cover letter; we will change the online submission form on your behalf. 4. As required by our policy on Data Availability, please ensure your manuscript or supplementary information includes the following:  A numbered table of all studies identified in the literature search, including those that were excluded from the analyses.   For every excluded study, the table should list the reason(s) for exclusion.   If any of the included studies are unpublished, include a link (URL) to the primary source or detailed information about how the content can be accessed.  A table of all data extracted from the primary research sources for the systematic review and/or meta-analysis. The table must include the following information for each study:  Name of data extractors and date of data extraction  Confirmation that the study was eligible to be included in the review.   All data extracted from each study for the reported systematic review and/or meta-analysis that would be needed to replicate your analyses.  If data or supporting information were obtained from another source (e.g. correspondence with the author of the original research article), please provide the source of data and dates on which the data/information were obtained by your research group.  If applicable for your analysis, a table showing the completed risk of bias and quality/certainty assessments for each study or outcome.  Please ensure this is provided for each domain or parameter assessed. For example, if you used the Cochrane risk-of-bias tool for randomized trials, provide answers to each of the signalling questions for each study. If you used GRADE to assess certainty of evidence, provide judgements about each of the quality of evidence factor. This should be provided for each outcome.   An explanation of how missing data were handled.   This information can be included in the main text, supplementary information, or relevant data repository. Please note that providing these underlying data is a requirement for publication in this journal, and if these data are not provided your manuscript might be rejected.  

Additional Editor Comments:

Dear Authors,

Thank you for submitting your manuscript titled “Predictors of maternal HIV acquisition during pregnancy and lactation in sub-Saharan Africa: a systematic review and narrative synthesis” for review. . Below are my comments that you might address or clarify, which I hope will assist you in enhancing the manuscript.

1. Methodological Concerns

1.1 Heterogeneity in Predictor Definitions

There is considerable variability in how key predictors (e.g., STIs, condom use, multiple partners) were defined across the included studies. The manuscript acknowledges this variability but does not address how it impacts the overall interpretation of results.

Consider providing a more robust discussion on how the varying definitions of predictors could have influenced your findings. A sensitivity analysis or stratified synthesis, where you group studies based on the homogeneity of predictor definitions, would add rigor to your approach.

1.2 Exclusion of Predictors Assessed in Fewer Than Four Studies

Excluding predictors evaluated in fewer than four studies may result in the omission of important but less-studied risk factors.

Recommendation: Provide a brief discussion of the predictors that were excluded and highlight any that may be emerging as important in other contexts or regions. If possible, explore these predictors narratively, even if formal synthesis was not feasible.

1.3 Limited Consideration of Confounders

The manuscript lacks adequate discussion of how confounding variables (e.g., socioeconomic status, healthcare access) were addressed in the included studies.

Add a section that categorizes how well the included studies adjusted for key confounders. Where confounding was not properly addressed, note how this limitation could affect the interpretation of the findings. This is particularly important for predictors like age and multiple partners, where social and healthcare access factors might play a significant role.

2. Presentation of Data and Interpretation of Results

2.1 Confidence Intervals and Precision

Many associations are reported without sufficient emphasis on the confidence limit ratios (CLRs) or the wide confidence intervals seen in certain predictors, which could indicate high imprecision.

Place greater emphasis on the interpretation of these confidence intervals and discuss how imprecision affects the reliability of the findings. If the CLRs indicate high variability, it would be important to highlight this as a limitation in your results.

2.2 Interpretation of Unexpected Findings (e.g., Condom Use)

The finding that consistent condom use is associated with a higher risk of HIV acquisition is counterintuitive and requires a deeper exploration of potential biases or alternative explanations.

Acknowledge the potential for reporting biases or misclassification of condom use. It is also important to discuss whether consistent condom use is related to other confounding variables, such as higher sexual activity, which may explain the increased risk observed.

2.3 Underrepresentation of Regional Differences

The exclusion of studies from Western Africa and non-English studies raises concerns about generalizability.

Include a more detailed discussion of how this exclusion impacts the conclusions, particularly given the diverse healthcare and cultural contexts in sub-Saharan Africa

3. Discussion and Implications for Policy and Practice

3.1 Lack of Practical Implications for Healthcare Providers

The manuscript focuses on identifying predictors but lacks specific recommendations for how these findings can be used in clinical practice or policy development.

Strengthen the discussion by providing concrete suggestions for how healthcare providers in sub-Saharan Africa can use these predictors in clinical decision-making. For instance, how can HIV risk screening be adapted based on the predictors identified in your review? What are the implications for PrEP implementation or antenatal care protocols?

3.2 Policy Recommendations

The current policy implications are underdeveloped and lack specificity.

Provide a more detailed exploration of how your findings align with or challenge current WHO recommendations. You could also offer suggestions for regional or national health policies that could be adapted to better address the identified predictors of HIV acquisition.

4. Clarity and Organization

4.1 Consistency in Data Presentation

Data presentation is sometimes inconsistent, particularly in terms of reporting confidence intervals and discussing effect sizes.

Ensure that all data, especially confidence intervals, are consistently reported. This will help readers more easily assess the strength and precision of your findings.

5.1 Expansion of Limitations and Future Research Directions

The limitations section is currently underdeveloped, particularly around the methodological choices made (e.g., vote counting, reliance on self-reported data).

Expand the limitations section to address the methodological constraints, including the exclusion of certain predictors and the impact of self-reported data. Additionally, provide more specific future research directions, such as focusing on under-researched predictors or conducting more region-specific analyses.

6. Engagement with Current Literature

6.1 Integration with Recent Studies and Trends

The manuscript does not sufficiently engage with recent advancements in HIV prevention, such as the growing use of PrEP or novel behavioral interventions.

Incorporate a discussion on how PrEP, telehealth, and other modern interventions could interact with the predictors you identified. This would make the paper more relevant to current clinical and policy debates.

Conclusion:

I look forward to reviewing a revised version.

Best regards,

Reviewers' comments:

Reviewer's Responses to Questions

**Comments to the Author**

1. Is the manuscript technically sound, and do the data support the conclusions?

Reviewer #1: Yes

Reviewer #2: Yes

2. Has the statistical analysis been performed appropriately and rigorously? 

Reviewer #1: N/A

Reviewer #2: Yes

3. Have the authors made all data underlying the findings in their manuscript fully available?

Reviewer #1: Yes

Reviewer #2: Yes

4. Is the manuscript presented in an intelligible fashion and written in standard English?

Reviewer #1: Yes

Reviewer #2: Yes

5. Review Comments to the Author

Reviewer #1: This is a well-written systematic meta-review of literature from SSA regarding risk factors for HIV acquisition among pregnant and lactating women. It includes 3+ decades of literature and identifies a common set of risk factors associated with HIV acquisition during pregnancy and lactation. The authors undergo a robust critique of the included manuscripts and draw measured conclusions about the common factors identified.

Background:

Would include brief discussion of why women are at particular risk of HIV during pregnancy and lactation. (social/family factors, biology/immunologic factors) Mofesen 2018 might be helpful here.

Would also include a brief context of the epidemiology of HIV acquisition among PLW in SSA.

Methods:

Initially your wide date range for inclusion struck me as a limitation—how relevant could studies of HIV acquisition risk in the late 80’s be for this analysis?—but ultimately the data convinced me in that the same themes emerged throughout the 30+ years you included. I think you would draw the same conclusions from a more narrow, homogeneous time period though I agree that the consistency of the themes over that period actually end up adding validity to your conclusion.

Results:

The tables are clear and helpful.

Discussion:

458-60: It was unclear to me what you meant by this sentence.

466: Can you say more here about current performance of the risk assessment tools you mention—either specifically more about the Kenya tool or provide an additional example? The discussion of risk factors is nuanced but I am left without a clear sense of how these findings could augment current practice.

Reviewer #2: The authors did an excellent job of synthesizing the diverse and broad range of published individual risk factors for acquiring HIV during pregnancy and lactation, using the standard PRISMA guidelines for systematic review. The authors struck a good balance of providing the diversity of various risk factor definitions and risk periods and distilling these definitions to meaningful categories. In discrete datasets, the analytic approach is generally to identify risk factors significantly associated with incident HIV and then to perform a multiple regression analysis to identify the subset of factors that remain independently associated with HIV incidence, since risk factors can confound the individual associations. The authors acknowledge this possibility in the Discussion (lines 455-457) by noting that the association between condom use and HIV infection could be confounded by other risk factors such as multiple sexual partners or having a partner living with untreated HIV. The diversity of type and definition of predictors precluded meta-analysis; however, I think that the discussion could have benefitted from a more robust discussion on the interplay between different risk factors and how to deal analytically with confounders in a risk scoring system- it's not just an issue of understanding that association does not mean causation (lines 491-2). The authors mention that some risk factors are self-reported. People might not be completely transparent about socially stigmatizing risks, and as noted, risks can be transitory. There has been a recent shift in prevention to deemphasize risk screening tools and instead provide prevention methods based on self-perceived risk, which the authors note was consistently associated with maternal HIV acquisition. Table 2 shows a moderate to high risk of bias in the publications. Given these limitations, in the discussion, I encourage you to reflect on the advantages/ disadvantages of a universal vs risk-based approach as trialed in the PrIMA study by Kinuthia, J., et al. (2023). "Risk-based versus universal PrEP delivery during pregnancy: a cluster randomized trial in Western Kenya from 2018 to 2019." J Int AIDS Soc 26(2): e26061. I realize this universal offer approach undermines the premise of this systematic review; however, it is a pragmatic alternative approach that could be acknowledged in the limitations. Another angle on the differences between studies is who was asking the risk questions and where. Participants might be more willing to be accurate (or honest) in certain circumstances (private location) and to certain interviewers. Specific minor points are that the legend in Figure 2 (page 50-54) are small and difficult to read, which makes it difficult to grasp what variable the forest plots are trying to show. You had a clear rationale for selecting risk factors that were assessed in at least 4 publications. That's an okay rationale - you had to limit in some way: however, I might have weighted the large studies (Dihn, Humphrey, Mussa, Woldesenbet) with many seroconversions that assessed risk factors (such as age difference) that didn't quite make the 4 study mark. (again, a minor issue)

6. PLOS authors have the option to publish the peer review history of their article (what does this mean?). If published, this will include your full peer review and any attached files.

Reviewer #1: No

Reviewer #2: **Yes: **Wm. Perry Killam

---

## [Author Response · Author response to Decision Letter 0]

22 Oct 2024

Editor Comments

1. Methodological Concerns

1.1 Heterogeneity in Predictor Definitions

There is considerable variability in how key predictors (e.g., STIs, condom use, multiple partners) were defined across the included studies. The manuscript acknowledges this variability but does not address how it impacts the overall interpretation of results. Consider providing a more robust discussion on how the varying definitions of predictors could have influenced your findings. A sensitivity analysis or stratified synthesis, where you group studies based on the homogeneity of predictor definitions, would add rigor to your approach.

Variability in how—and when—predictors were measured is a key finding of this systematic review. While we agree with the editor regarding the value of a more formal investigation into how the definitions of a predictor influenced the direction or magnitude of association, this was not possible given the heterogeneity of predictor definitions and the total number of publications included in this review (i.e. age was operationalized in 14 different ways across the 20 publications that assessed maternal age). Thus, we report how each predictor was measured in every study in Table S2 and focused on identifying overall trends using vote counting. When possible, we distilled varied definitions into common contrasts to provide a better sense of the magnitude and range of associations between each predictor and risk of maternal HIV acquisition (Table 4 and Figures 2-6).

1.2 Exclusion of Predictors Assessed in Fewer Than Four Studies

Excluding predictors evaluated in fewer than four studies may result in the omission of important but less-studied risk factors. I recommend providing a brief discussion of the predictors that were excluded and highlight any that may be emerging as important in other contexts or regions. If possible, explore these predictors narratively, even if formal synthesis was not feasible.

Table 3 provides a summary of every predictor evaluated by included publications. We revised our Discussion to summarize potentially important risk factors not included in our analysis (i.e. those identified as important predictors of HIV acquisition among women of reproductive age in SSA; lines: 492-502).

1.3 Limited Consideration of Confounders

The manuscript lacks adequate discussion of how confounding variables (e.g., socioeconomic status, healthcare access) were addressed in the included studies.

Add a section that categorizes how well the included studies adjusted for key confounders. Where confounding was not properly addressed, note how this limitation could affect the interpretation of the findings. This is particularly important for predictors like age and multiple partners, where social and healthcare access factors might play a significant role.

Adjustment for confounding variables is relevant when the goal of an analysis is to obtain unbiased estimates of effect that reflect the causal relationship between an exposure and an outcome (Hernan and Robins, 2020). This is distinct from the goal of our analysis, which was to identify predictors of maternal HIV acquisition (lines: 83-84). While multivariable models can be used to identify predictors that are independently associated with the outcome, we focused on crude estimates of association because adjusted estimates from non-causal models are difficult to interpret and are often not comparable across studies. This is stated in the second paragraph of the Methods (lines: 105-106; references 14 and 15). In our Discussion, we acknowledge that the unadjusted associations summarized in our manuscript may not correspond to causal effects and thus should not be used to identify targets for interventions (lines: 520-526).

Hernán MA, Robins JM (2020). Causal Inference: What If. Boca Raton: Chapman & Hall/CRC.

2. Presentation of Data and Interpretation of Results

2.1 Confidence Intervals and Precision

Many associations are reported without sufficient emphasis on the confidence limit ratios (CLRs) or the wide confidence intervals seen in certain predictors, which could indicate high imprecision. Place greater emphasis on the interpretation of these confidence intervals and discuss how imprecision affects the reliability of the findings. If the CLRs indicate high variability, it would be important to highlight this as a limitation in your results.

In the fifth paragraph of the Results, we state that estimated associations were imprecise, as illustrated by median CLRs that were greater than 1 (lines: 207-208). We revised the legend for Figures 2-6 to make it clearer that the size of the point estimate symbol and opacity of the confidence interval line are directly proportional to the CLR. We revised our Discussion to acknowledge that imprecise estimates of association are difficult to rely on, particularly in isolation (lines: 443-445). By summarizing across studies, however, we were able to look at the directional consistency of estimated associations across diverse populations and time. We consider this a strength of our approach. While trends based on imprecise estimates should be interpreted cautiously, consistency in the direction of association provides greater confidence in the relationship between the predictor and risk of maternal HIV acquisition. We added this to the discussion (lines: 461-463).

2.2 Interpretation of Unexpected Findings (e.g., Condom Use)

The finding that consistent condom use is associated with a higher risk of HIV acquisition is counterintuitive and requires a deeper exploration of potential biases or alternative explanations. Acknowledge the potential for reporting biases or misclassification of condom use. It is also important to discuss whether consistent condom use is related to other confounding variables, such as higher sexual activity, which may explain the increased risk observed.

We discuss how the observed association between HIV acquisition and condom use may reflect the fact that, in SSA, condoms are typically used in the context of relationships perceived as carrying higher HIV acquisition risk (lines: 456-458). We agree that misclassification is another potential explanation and have added this to the discussion (line: 455-456).

2.3 Underrepresentation of Regional Differences

The exclusion of studies from Western Africa and non-English studies raises concerns about generalizability. Include a more detailed discussion of how this exclusion impacts the conclusions, particularly given the diverse healthcare and cultural contexts in sub-Saharan Africa

This is an important point, and we agree that our results may not generalize to countries in Africa with a concentrated HIV epidemic. We have revised our discussion accordingly (lines: 463-469).

3. Discussion and Implications for Policy and Practice

3.1 Lack of Practical Implications for Healthcare Providers

The manuscript focuses on identifying predictors but lacks specific recommendations for how these findings can be used in clinical practice or policy development. Strengthen the discussion by providing concrete suggestions for how healthcare providers in sub-Saharan Africa can use these predictors in clinical decision-making. For instance, how can HIV risk screening be adapted based on the predictors identified in your review? What are the implications for PrEP implementation or antenatal care protocols?

As stated in the Introduction, the objective of this review was to identify and characterize common predictors of maternal HIV acquisition in SSA (lines: 83-84). Alone, our results cannot be used to guide clinical decision-making because unvalidated tools to screen patients for PrEP eligibility may be insufficient for measuring risk (see: Steyerberg and colleagues, 2010). However, as we note in the Discussion, our results can be used to support the development and refinement of tools to assess HIV risk among PLW in SSA (lines: 486-490). 

A common precursor to prediction modeling is a predictor finding study. Such studies don’t aim to develop a full prediction model, but instead seek to identify predictors that could augment existing tools. By synthesizing results of individual predictor finding studies, this review provides stronger evidence for what predictors should be considered in future predictive algorithm development. This approach is recommended by the PROGRESS Group to enhance the impact and generalizability of predictive modeling. We added a reference to this recommendation to the manuscript (line: 85 and 463, reference 12).

Steyerberg EW, Vickers AJ, Cook NR, Gerds T, Gonen M, Obuchowski N, Pencina MJ, Kattan MW. Assessing the performance of prediction models: a framework for traditional and novel measures. Epidemiology. 2010 Jan;21(1):128-38.

3.2 Policy Recommendations

The current policy implications are underdeveloped and lack specificity. Provide a more detailed exploration of how your findings align with or challenge current WHO recommendations. You could also offer suggestions for regional or national health policies that could be adapted to better address the identified predictors of HIV acquisition.

As we note in the Discussion (lines: 520-526), results from this review cannot be used to identify targets for intervention because they reflect crude (unadjusted) estimates of association. We are therefore unable to suggest regional or national health policies that could be adapted to address identified predictors of HIV acquisition. Our recommendation that results be used to develop and refine HIV risk assessment tools, however, clearly maps to global guidance on PrEP implementation and prediction model development (see references 3, 4, 11, and 12).

4. Clarity and Organization

4.1 Consistency in Data Presentation

Data presentation is sometimes inconsistent, particularly in terms of reporting confidence intervals and discussing effect sizes. Ensure that all data, especially confidence intervals, are consistently reported. This will help readers more easily assess the strength and precision of your findings.

Table 3 provides the median and range of relative risks and relative rates, and the median and range of the CLR, for every predictor where common contrasts could be generated. We opted to report the CLR instead of confidence intervals because the CLR is a standardized metric of precision (see references 17 and 18). The results presented in Table 3 map to our narrative descriptions of these results in subsequent sections. We revised the legend for Figures 2-6 to make it clearer that the size of the point estimate symbol and opacity of the confidence interval lines are directly proportional to the CLR.

5.1 Expansion of Limitations and Future Research Directions

The limitations section is currently underdeveloped, particularly around the methodological choices made (e.g., vote counting, reliance on self-reported data). Expand the limitations section to address the methodological constraints, including the exclusion of certain predictors and the impact of self-reported data. Additionally, provide more specific future research directions, such as focusing on under-researched predictors or conducting more region-specific analyses.

We do not consider the use of self-reported data on predictors to be a limitation because this approach emulates how such data would be collected (and predictors used) in real world clinical settings. We’ve elaborated on the following limitations in the Discussion (lines: 463-469 and 531-548): 

- Questionable generalizability

- Vote counting

- Exclusion of conference abstracts

- Reporting biases

- Loss of information when dichotomizing variables

We also revised our discussion to more clearly state future research directions related to:

- Identifying predictors of maternal HIV acquisition in west/central Africa (lines: 469-471).

- Developing and implementing tools to assess HIV risk (lines: 486-489).

- Understanding how partner-level attributes shape maternal HIV risk (lines: 502-504)

- Developing HIV risk trajectories over pregnancy and lactation to inform dynamic HIV prevention approaches (lines: 515-518).

6. Engagement with Current Literature

6.1 Integration with Recent Studies and Trends

The manuscript does not sufficiently engage with recent advancements in HIV prevention, such as the growing use of PrEP or novel behavioral interventions. Incorporate a discussion on how PrEP, telehealth, and other modern interventions could interact with the predictors you identified. This would make the paper more relevant to current clinical and policy debates.

As noted by the editor, our findings are primarily derived from data collected prior to the widespread implementation of combination HIV services. However, as we note in the Discussion, we believe that the strong biological and behavioral links between the predictors identified in this review and HIV acquisition across several decades suggest their continued relevance, even as the epidemic landscape evolves (lines: 465-469).

their continued relevance, even as the epidemic landscape evolves (lines: 465-469). 

Reviewer #1 Comments

This is a well-written systematic meta-review of literature from SSA regarding risk factors for HIV acquisition among pregnant and lactating women. It includes 3+ decades of literature and identifies a common set of risk factors associated with HIV acquisition during pregnancy and lactation. The authors undergo a robust critique of the included manuscripts and draw measured conclusions about the common factors identified.

Thank you for this positive feedback.

Background:

Would include brief discussion of why women are at particular risk of HIV during pregnancy and lactation. (social/family factors, biology/immunologic factors) Mofesen 2018 might be helpful here. 

We added a brief discussion in the second Introduction paragraph on the biological and behavioral changes in pregnancy that may increase risk of HIV acquisition (lines: 71-75), citing Mofenson 2018 (reference 7). 

Would also include a brief context of the epidemiology of HIV acquisition among PLW in SSA.

As we note in the Introduction, an estimated 120,000 pregnant and/or lactating women acquired HIV across the 21 UNAIDS focus countries in 2020 (lines: 58-59). By characterizing common predictors of maternal HIV acquisition in SSA (our objective, stated on lines: 83-84), our results contextualize the high average risk of HIV acquisition among PLW estimated in prior work.

Methods:

Initially your wide date range for inclusion struck me as a limitation—how relevant could studies of HIV acquisition risk in the late 80’s be for this analysis?—but ultimately the data convinced me in that the same themes emerged throughout the 30+ years you included. I think you would draw the same conclusions from a more narrow, homogeneous time period though I agree that the consistency of the themes over that period actually end up adding validity to your conclusion.

We agree that consistent trends across time and populations provides greater confidence in the relationship between the predictor and risk of maternal HIV acquisition. We added this point to our discussion (lines: 461-463).

Results:

The tables are clear and helpful.

Thank you.

Discussion:

458-60: It was unclear to me what you meant by this sentence.

HIV risk is shaped by the combination of individual- and partner attributes, some of which may be influenced or proxied by community-level drivers or measures. While individual- and partner-level attributes were evaluated by included publications, no study included in this review considered whether community-level factors that proxy HIV exposure risk (e.g., HIV incidence, prevalence of viremia, etc.) were associated with HIV acquisition among PLW. As subnational prevalence of viremia was the single most important predictor of recent HIV infection among women aged 15-49 in 15 African countries (reference 66), and HIV incidence is included as part of UNAIDS risk classification algorithm (reference 11) we felt this was an important factor to highlight. We have revised our discussion for clarity (lines: 492-502).

466: Can you say more here about current performance of the risk assessment tools you mention—either spe

---

## [Decision Letter · Decision Letter 1]

18 Nov 2024

Predictors of maternal HIV acquisition during pregnancy and lactation in sub-Saharan Africa:  a systematic review and narrative synthesis

PONE-D-24-20929R1

Dear Dr. Graybill,

We’re pleased to inform you that your manuscript has been judged scientifically suitable for publication and will be formally accepted for publication once it meets all outstanding technical requirements.

Kind regards,

Garumma Tolu Feyissa, PhD

Academic Editor

PLOS ONE

Additional Editor Comments (optional):

Reviewers' comments:

Reviewer's Responses to Questions

**Comments to the Author**

1. If the authors have adequately addressed your comments raised in a previous round of review and you feel that this manuscript is now acceptable for publication, you may indicate that here to bypass the “Comments to the Author” section, enter your conflict of interest statement in the “Confidential to Editor” section, and submit your "Accept" recommendation.

Reviewer #1: All comments have been addressed

Reviewer #2: All comments have been addressed

2. Is the manuscript technically sound, and do the data support the conclusions?

Reviewer #1: Yes

Reviewer #2: Yes

3. Has the statistical analysis been performed appropriately and rigorously? 

Reviewer #1: N/A

Reviewer #2: Yes

4. Have the authors made all data underlying the findings in their manuscript fully available?

Reviewer #1: Yes

Reviewer #2: Yes

5. Is the manuscript presented in an intelligible fashion and written in standard English?

Reviewer #1: Yes

Reviewer #2: Yes

6. Review Comments to the Author

Reviewer #1: Appreciate your responses to my particular comments, as well as the comments by the other reviewer. This is a comprehensive meta-analysis of literature, though agree with the other review that an update will be imminently needed in light of some of the recent advances in PrEP provision among PBFW in sub-Saharan Africa.

Reviewer #2: Thank you for responding to our comments. I think this is now a stronger paper, and I recommend that PLOS One accept it.

7. PLOS authors have the option to publish the peer review history of their article (what does this mean?). If published, this will include your full peer review and any attached files.

Reviewer #1: No

Reviewer #2: **Yes: **Wm. Perry Killam

---

## [Editor Report · Acceptance letter]

19 Nov 2024

PONE-D-24-20929R1 

PLOS ONE

Dear Dr. Graybill, 

I'm pleased to inform you that your manuscript has been deemed suitable for publication in PLOS ONE. Congratulations! Your manuscript is now being handed over to our production team.

Kind regards, 

on behalf of

Dr. Garumma Tolu Feyissa 

Academic Editor

PLOS ONE